# Single cell sequencing identifies clonally expanded synovial CD4$^+$ T$_{PH}$ cells expressing GPR56 in rheumatoid arthritis

Alexandra Argyriou [1,4], Marc H. Wadsworth II [2,4], Adrian Lendvai[1], Stephen M. Christensen [2], Aase H. Hensvold[1,3], Christina Gerstner[1], Annika van Vollenhoven[1], Kellie Kravarik [2], Aaron Winkler[2], Vivianne Malmström [1] & Karine Chemin[1✉]

Rheumatoid arthritis (RA) is an autoimmune disease affecting synovial joints where different CD4$^+$ T cell subsets may contribute to pathology. Here, we perform single cell sequencing on synovial CD4$^+$ T cells from anti-citrullinated protein antibodies (ACPA)+ and ACPA- RA patients and identify two peripheral helper T cell (T$_{PH}$) states and a cytotoxic CD4$^+$ T cell subset. We show that the adhesion G-protein coupled receptor 56 (GPR56) delineates synovial CXCL13$^{high}$ T$_{PH}$ CD4$^+$ T cells expressing LAG-3 and the tissue-resident memory receptors CXCR6 and CD69. In ACPA- SF, T$_{PH}$ cells display lower levels of GPR56 and LAG-3. Further, most expanded T cell clones in the joint are within CXCL13$^{high}$ T$_{PH}$ CD4$^+$ T cells. Finally, RNA-velocity analyses suggest a common differentiation pathway between the two T$_{PH}$ clusters and effector CD4$^+$ T cells. Our study provides comprehensive immunoprofiling of the synovial CD4$^+$ T cell subsets in ACPA+ and ACPA- RA.

[1] Division of Rheumatology, Center for Molecular Medicine, Department of Medicine, Solna, Karolinska Institutet, Karolinska University Hospital, Stockholm, Sweden. [2] Inflammation & Immunology Research Unit, Pfizer Inc., Cambridge, MA 02139, USA. [3] Center for Rheumatology, Academic Specialist Center, Stockholm Health Region, Stockholm, Sweden. [4]These authors contributed equally: Alexandra Argyriou, Marc H. Wadsworth II. ✉email: karine.chemin@ki.se

Rheumatoid arthritis (RA), which affects around 1% of the global population, is an autoimmune disease characterized by articular cartilage and bone erosion leading to physical disability, pain, and decreased quality of life. Joint damage is particularly exacerbated in the subset of RA patients who present with antibodies against citrullinated proteins (ACPA)[1,2]. Although the use of biologics has revolutionized the treatment of RA, 30–40% of patients fail to respond to treatment stressing the need for novel therapeutic targets[3]. The activity of CD4+ T cells in RA pathogenesis is evidenced by the HLA-DRB1 association with ACPA+ RA[4]. Expanded patient-specific T-cell clones have been identified in the synovium and synovial fluid (SF) of RA patients using bulk TCRβ repertoire analysis[5] but without any connection to their effector function or antigen reactivities. Over the years, several CD4+ T-cell subsets, including Th1 and Th17 cells have been described in the synovial joints of RA patients[6]. Recently, PD-1$^{high}$ MHC-II+ (major histocompatibility complex class II) peripheral helper T cells ($T_{PH}$) were identified in the synovial fluid and tissue of seropositive RA patients[7–9] where they are proposed to facilitate B cell recruitment and activation through CXCL13 and IL-21 production; however, there is still no satisfying marker that defines the $T_{PH}$ subset. Furthermore, it is also unclear how this population is generated in the synovial joint. We have recently identified CD4+ T cells with cytotoxic features in SF of RA patients carrying the *PTPN22* 1858T risk allele variant[10]. This finding, as well as previous studies reporting expanded cytotoxic CD4+ CD28$^{null}$ T cells in the blood of RA patients[11,12], suggests that cytotoxic CD4+ T cells might contribute to RA. However, a transcriptomic and clonality analysis of cytotoxic CD4+ T cells in the context of previously described CD4+ T-cell subsets, is still missing. The identification of T-cell effector functions is deemed important since it can lead to new therapeutic strategies.

Here, we performed 5′single-cell sequencing in combination with TCRαβ sequencing on peripheral blood (PB) and synovial fluid (SF) from ACPA+ and ACPA− RA patients to investigate CD4+ T-cell subsets. We demonstrate that clonal expansions are prominent amongst CXCL13$^{high}$ $T_{PH}$, Treg, effector and cytotoxic CD4+ T cells in RA SF. We also describe two $T_{PH}$ CD4+ T-cell clusters expressing different levels of *CXCL13*. Finally, we find that $T_{PH}$ cells express the G-protein-coupled receptor GPR56, the inhibitory receptor LAG-3, and the tissue-resident memory ($T_{RM}$) receptors CXCR6 and CD69 implicating that these T cells are maintained in the ACPA+ RA synovial joint. Our data provide a comprehensive immunoprofiling analysis of pathogenic CD4+ T cells at the site of inflammation in ACPA− and ACPA+ RA.

## Results

**Cytotoxic CD4+ T cells are enriched in ACPA+ RA synovial fluid.** CD4+ T cells with cytotoxic function share common functional characteristics with NK and CD8+ T cells, including the expression of cytolytic proteins PRF1 (perforin-1), GZMB (granzyme B)[13] and NKG7 (natural killer cell granule protein 7)[14]. Both transcription factors Eomes[15,16] and Hobit[17] are expressed in circulating human cytotoxic T cells. GPR56 is an adhesion G-protein-coupled receptor encoded by the *ADGRG1* gene expressed on human circulating NK, CD8+, and CD4+ cytotoxic cells[18]. Synovial fluid mononuclear cells (SFMC) from ACPA− ($n = 9$) and ACPA+ ($n = 12$) RA patients were screened for the expression of these cytotoxic effector molecules and transcription factors in both CD4+ and CD8+ T cells by flow cytometry (Fig. 1, Supplementary Figs. 1a and 2, and Supplementary Data 1). ACPA+ SFMC presented with a significantly increased frequency of GZMB+ PRF1+, Hobit+, NKG7$^{high}$, and GPR56+ CD4+ cells (Fig. 1a, b). No significant difference was observed for expression of GZMA and

Eomes. Among CD8+ T cells, Hobit expression was increased in ACPA+ SFMC and a similar trend was observed for the expression of GZMA, GZMB, and NKG7 without reaching statistical significance (Supplementary Fig. 2). No significant difference was observed in the expression of PRF1 and GPR56 on CD8+ T cells in ACPA+ versus ACPA− RA. The frequency of GZMB+ PRF1+ CD4+ T cells in SF positively correlated with CCP (cyclic citrullinated peptide) positivity (Fig. 1c). In ACPA+ synovial fluid, GPR56 identified a distinct CD4+ T-cell population with a frequency that also correlated with CCP positivity (Fig. 1d).

**Single-cell sequencing map of CD4+ T-cell subsets in RA.** To further characterize cytotoxic CD4+ T cells in SFMC, we performed 10X 5′single-cell sequencing in combination with TCRαβ sequencing on purified CD4+ T cells from paired SFMC and peripheral blood mononuclear cells (PBMC) from 11 ACPA+ RA patients and 4 ACPA− patients (Fig. 2a and Supplementary Data 1). After quality control (Supplementary Figs. 3 and 4a), we obtained transcriptomic data from a total of 166,944 cells. Unsupervised analysis of this transcriptomic data generated 12 total clusters that were annotated based on both known cell-type markers and enrichment for cell-type-specific gene modules (Fig. 2b, c, Supplementary Figs. 4b, c and 5a, b, and Supplementary Data 2 and 3). Differentially-expressed genes (DEGs) were also compared to clusters described in PB of healthy donors[19] and the Azimuth reference single-cell mapping database[20,21] using the EnrichR interface[22] (Supplementary Fig. 5c). Together, using this approach we identified a naive CD4+ T-cell cluster (cluster 1) expressing high levels of *SELL*, *CCR7*, and *IL7R*[23], central memory CD4+ T cells (cluster 3) expressing *LTB* and *KLF2*; effector CD4+ T cells (cluster 4) expressing *KLRB1* and *GZMA*; Tregs (cluster 5) expressing *FOXP3* and *IL2Ra*[24]; proliferating CD4+ T cells (cluster 11) expressing *MKI67* and *STMN1*; cytotoxic CD4+ T cells (cluster 6) expressing cytotoxicity related genes including *NKG7*, *GZMH*, *PRF1* and *ZNF683* (Hobit) among others; activated CD4+ T cells (cluster 12) expressing CD38 and HLA-DR; SESN3$^{high}$ memory/effector CD4+ T cells (cluster 7) and EGR1$^{high}$ naive CD4+ T cells (cluster 10). Interestingly, SESN3$^{high}$ CD4+ T cells presented a cytokine signature matching with IL-2, TNF, or IL-6 signatures depending on the comparative datasets (Supplementary Fig. 5c). Cells from cluster nine expressed mitochondrial encoding genes and were named Humanin CD4+ T cells as previously described in psoriatic arthritis patients[25]. Finally, based on the expression of *CXCL13, MAF, TOX, PRDM1, TIGIT* and *PDCD1*[7,9,26], two $T_{PH}$ cell clusters were subclassified as CXCL13$^{high}$ $T_{PH}$ (cluster 2) and CXCL13$^{low}$ $T_{PH}$ (cluster 8). We further conducted our analysis on paired PB and SF compartment from four ACPA− and four ACPA+ RA patients (Fig. 2b, see "Single-cell sequencing of RA samples" in "Methods" and Supplementary Fig. 3). As expected, naive and central memory CD4+ T-cell frequencies were decreased in SF as compared to PB (Fig. 2d). Humanin and EGR1$^{high}$ CD4+ T-cell cluster frequencies were also decreased in SF. The frequency of Tregs was increased in SF as previously described by us and others[27,28]. Similarly, CXCL13$^{high}$ $T_{PH}$, CXCL13$^{low}$ $T_{PH}$, effector CD4+, proliferating CD4+, and activated CD4+ T-cell clusters frequencies were all increased in SF as compared to PB. Of note, the cytotoxic CD4+ T-cell cluster was equally frequent in PB and SF. These differences remain significant even when exclusively looking at the memory T-cell compartment (Supplementary Fig. 6a). No significant differences were observed between ACPA− and ACPA+ patients (Supplementary Fig. 6b, c). Of note, the frequency of the different clusters was not significantly different in patients undergoing corticosteroid, methotrexate, or biologics treatments at

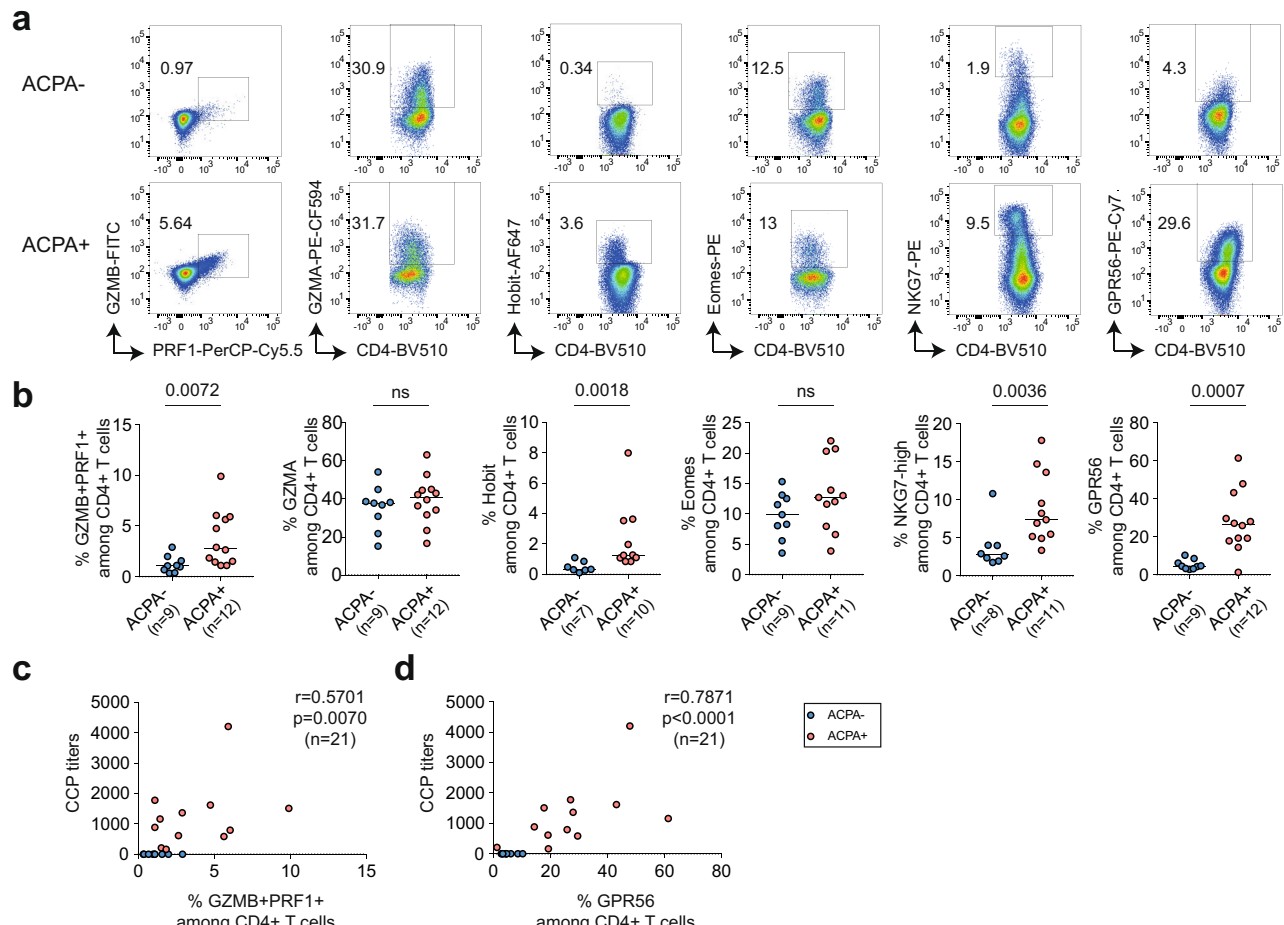

**Fig. 1 Cytotoxic CD4$^+$ T-cell frequency in synovial fluid of ACPA− and ACPA+ RA patients. a** Representative flow cytometry dot plot staining of effector molecules, receptors and transcription factors associated with cytotoxic functions in CD4$^+$ T cells from synovial fluid (SF) from ACPA− (upper panel) and ACPA+ (lower panel) RA patients, quantified in **b**, (ACPA−, $n = 7$–9) (ACPA+, $n = 10$–12). Line represents median, two-tailed Mann–Whitney $U$ test, $P = 0.0072$ (%GZMB$^+$PRF1$^+$), ns (%GZMA), $P = 0.0018$ (%Hobit), ns (%Eomes), $P = 0.0036$ (%NKG7-high), $P = 0.0007$ (%GPR56), ns: not significant. **c** Correlation between the frequency of GZMB$^+$ PRF1$^+$ in CD4$^+$ T cells in SF and the level of serum anti-CCP (cyclic citrullinated peptide) antibodies, $n = 21$, Spearman two-tailed test, $P = 0.0070$. **d** Correlation between the frequency of GPR56 in CD4$^+$ T cells in SF and the level of anti-CCP (cyclic citrullinated peptide) antibodies, $n = 21$, Spearman two-tailed test, $P < 0.0001$. **a**–**d** Data are from a pool of nine independent experiments where a circle is a single replicate. Blue dots indicate ACPA− RA SF and red dots indicate ACPA+ RA SF.

the time of sampling (Supplementary Fig. 7a). Hence, single-cell transcriptomic data identify several CD4$^+$ T-cell subsets, including a subset of cytotoxic CD4$^+$ T cells and two distinct T$_{PH}$ clusters in SF of RA patients.

**GPR56 delineates T$_{PH}$ CD4$^+$ T cells in ACPA+ RA synovial fluid**. Since GPR56 expression was upregulated on 26% of CD4$^+$ T cells in SF of ACPA+ RA patients (Fig. 1a, b), we further investigated *ADGRG1* (encoding GPR56) in the single-cell dataset. In PB, *ADGRG1* was mainly expressed by cytotoxic CD4$^+$ T cells (cluster 6), that are also characterized by the expression of *GZMA, GZMB, NKG7, PRF1* and *ZNF683* (Hobit) (Supplementary Fig. 8a–c). At the protein level, flow cytometry experiments validated that, in circulating CD4$^+$ T cells, GPR56 expression correlates with the expression of cytotoxic effector molecules PRF1, GZMB, GZMA, NKG7, and the transcription factors Eomes and Hobit (Supplementary Fig. 8d–f) confirming a previous report[18]. Still, a fraction of circulating GPR56$^+$ CD4$^+$ T cells expressed CXCL13 after CD3/CD28 stimulation (Supplementary Fig. 9). Interestingly, in SF, *ADGRG1* expression was mainly detected in CXCL13$^{high}$ T$_{PH}$ (cluster 2), CXCL13$^{low}$ T$_{PH}$

(cluster 8) and proliferating CD4$^+$ T cells (cluster 11) (Fig. 3a, Supplementary Fig. 10a–c). We further examined the expression of known T$_{PH}$-associated markers[7,9,26] and observed that *ADGRG1* expression overlapped with the expression of *PDCD1* (encoding PD-1), *CXCL13, HLA-DRB1, MAF, ICOS, PDRM1* (encoding BLIMP-1), and *TOX* in the two T$_{PH}$ states (Fig. 3a). At the RNA level, the two T$_{PH}$ clusters were characterized by different levels of *ADGRG1, PDCD1, CXCL13, HLA-DRB1, MAF*, and *PDRM1* (Supplementary Fig. 11a, b). In particular, in ACPA+ SF, *CXCL13, PDCD1*, and *ADGRG1* were increased in CXCL13$^{high}$ T$_{PH}$ whereas *PRDM1* was increased in CXCL13$^{low}$ T$_{PH}$ cells. The expression of *TOX* was not significantly different between the two subsets (Supplementary Fig. 11a). The same differences were also observed in T$_{PH}$ cells from ACPA− SF, in particular for *ADGRG1* and *PRDM1* (Supplementary Fig. 11b). Flow cytometry experiments confirmed previous reports[7,9] showing an increased frequency of PD-1$^{high}$ and PD-1 expression on CD4$^+$ T cells in ACPA+ as compared to ACPA− RA patients (Fig. 3b and Supplementary Fig. 12). In SF, GPR56 expression defines the subset of T$_{PH}$ cells (Fig. 3b) as illustrated by the increased expression of GPR56 on PD-1$^{high}$ as compared to PD-1$^{neg}$ CD4$^+$ T cells in both ACPA+ and ACPA− RA patients.

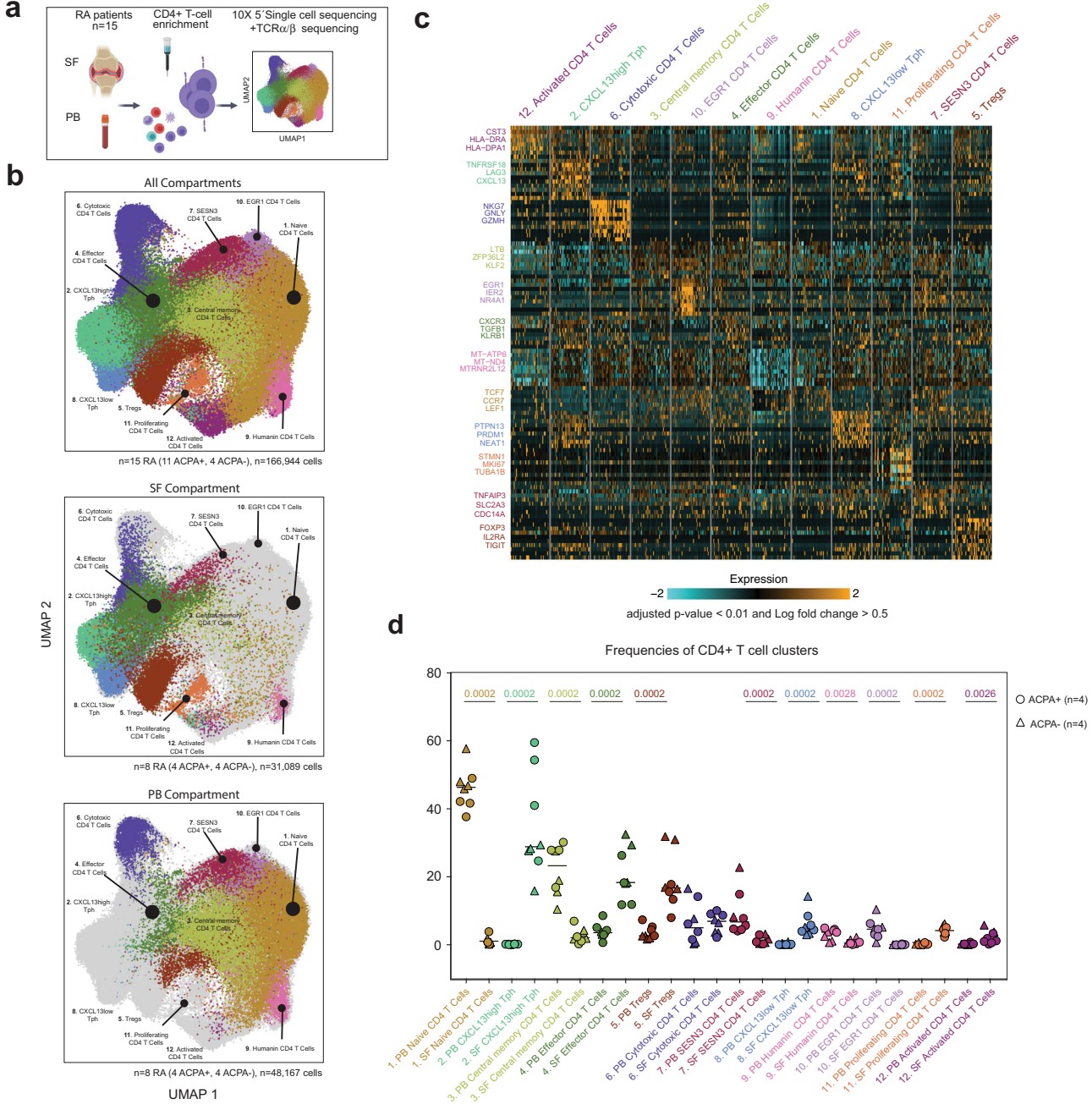

**Fig. 2 Single-cell RNA sequencing of CD4⁺ T cells from SF and PB of RA patients. a** Technical workflow including CD4⁺ T-cell enrichment and 10X 5′ single-cell sequencing coupled to TCRαβ sequencing on 15 paired PB and SF from RA patients (11 ACPA+, 4 ACPA−). **b** UMAP displaying 12 CD4⁺ T-cell clusters in combined SF and PB (upper panel, $n = 15$ RA (11 ACPA+, 4 ACPA−), $n = 166{,}944$ cells), SF (middle panel, $n = 8$ RA (4 ACPA+, 4 ACPA −), $n = 31{,}089$ cells) and PB (lower panel, $n = 8$ RA (4 ACPA+, 4 ACPA−), $n = 48{,}167$ cells). **c** Heatmap showing selected differentially-expressed genes (DEGs) in the different CD4⁺ T-cell clusters in combined PB and SF, $P < 0.01$, model-based analysis of single-cell transcriptomics (MAST). **d** Frequencies of CD4⁺ T-cell clusters in PB and SF from $n = 8$ RA patients (4 ACPA+, 4 ACPA−). Circle indicates ACPA+ RA patients, triangle indicates ACPA− patients. Line represents median, two-tailed Mann–Whitney $U$ test.

ACPA− T$_{PH}$ cells, however, expressed lower levels of GPR56 as their ACPA+ counterparts (Fig. 3b). Of note, these frequencies were not different depending on the treatment at the time of sampling (Supplementary Fig. 7b). We confirmed that the two T$_{PH}$ subsets could be distinguished based on different levels of GPR56 expression within PD-1$^{high}$ CD4⁺ T cells (Fig. 3c, gating strategy in Supplementary Fig. 1b). PD-1$^{high}$ GPR56⁺ CD4⁺ T cells expressed a higher level of MHC-II, PD-1, and CXCL13 as compared to PD-1$^{high}$ GPR56$^{low}$ CD4⁺ T cells (Fig. 3c–e). However, PD-1$^{high}$ GPR56⁺ CD4⁺ T cells also expressed higher

levels of BLIMP-1 although *PDRM1* gene expression was lower in this subset (Fig. 3e, lower panel and Supplementary Fig. 11a, b). While the frequency of PD-1$^{high}$ GPR56⁺ CD4⁺ T cells was lower in ACPA− SF (Fig. 3c, left panel), they also display increased CXCL13, MHC-II, and PD-1 expression as compared to their PD-1$^{high}$ GPR56$^{low}$ counterparts (Fig. 3c–e). GPR56 was not detected on cytotoxic CD4⁺ T cells in SF (Supplementary Fig. 10d, e) and the expression of *ADGRG1* was low in SF cytotoxic CD4⁺ T cells (Supplementary Fig. 10b, c). In NK cells, the expression of GPR56 is regulated by cleavage and endocytosis

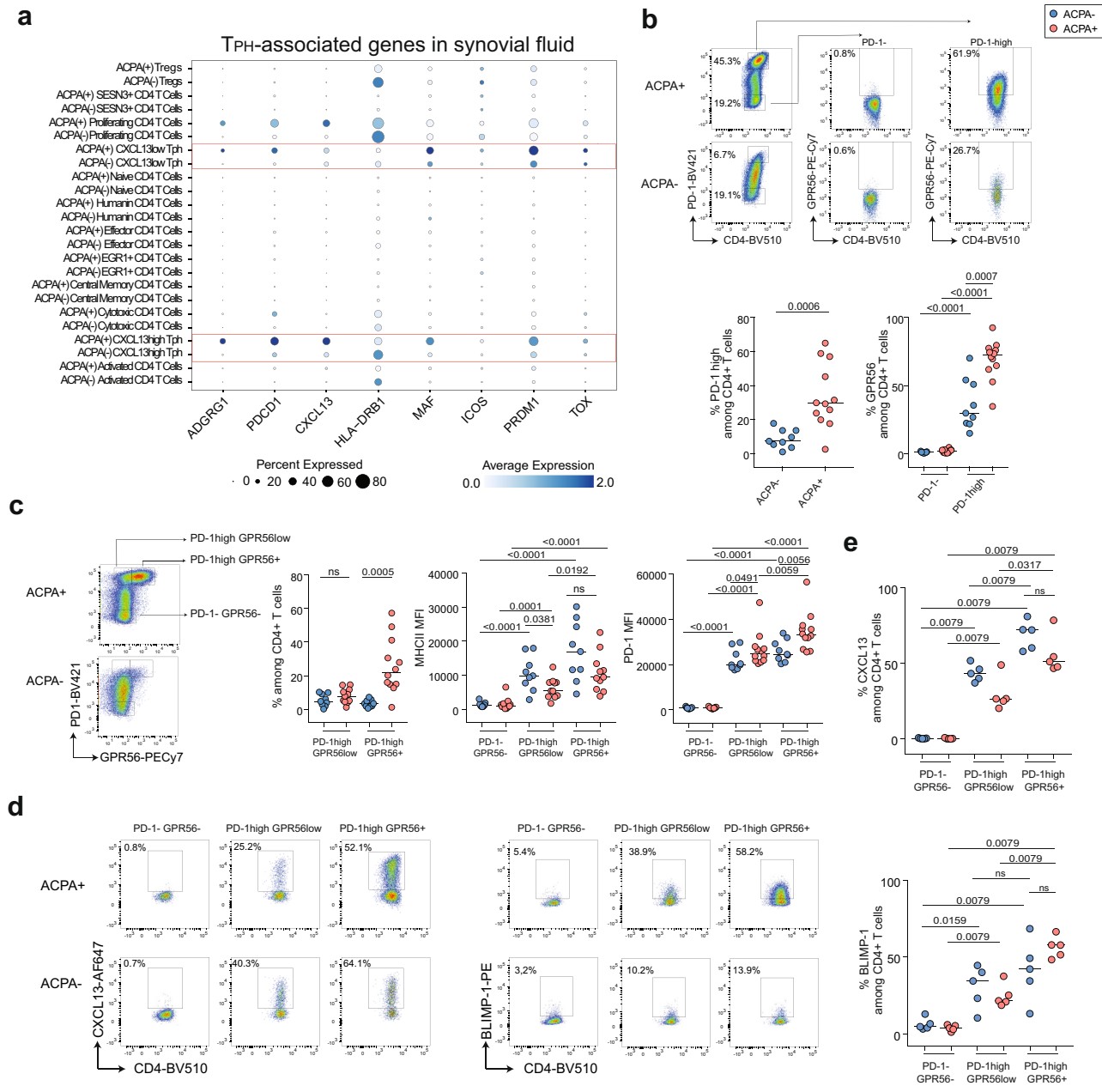

after activation[29] and a similar phenomenon could also contribute to the absence of GPR56 on cytotoxic CD4+ T cells in SF. We, therefore, investigated GPR56 expression after three hours of CD3/CD28 beads or PMA/ionomycin stimulation on PBMC and SFMC from RA patients. While three hours of CD3/CD28 stimulation had no effect on GPR56 expression, PMA/ionomycin induced downregulation of GPR56 surface expression on PB and SF CD4+ T cells (Supplementary Fig. 13). These experiments show that rapid GPR56 downregulation can also occur on GPR56+ CD4+ T cells through TCR-independent mechanisms. Altogether, these data indicate that, while GPR56 is mainly expressed on cytotoxic CD4+ T cells in PB, it also delineates the subset of PD-1high CXCL13high T_PH cells which is increased in ACPA+ RA SF.

**GPR56+ CD4+ T cells display distinct tissue-resident memory receptors**. Besides being an inhibitory receptor, PD-1 is also included in the signature associated to T-cell tissue residency[30]

and implicated in follicular helper T-cell maintenance in germinal centers[31]. We therefore assessed the expression of known tissue-resident memory (T_RM) T-cell receptors: ITGA1 (CD49a), CD69, CXCR6 but also CX3CR1 which is downregulated on T_RM cells[30]. We also evaluated the expression of the inhibitory receptor LAG-3 which is often co-expressed with PD-1 on tumor-infiltrating lymphocytes[32]. In ACPA+ SF, we found that LAG-3 was mainly expressed on T_PH cells (cluster 2 and 8) and proliferating T cells (cluster 11) (Fig. 4a). A similar tendency was also observed for the expression of CXCR6 in SF. Since we previously observed that GPR56 delineates the subset of T_PH cells, we assessed the expression of these receptors in the context of GPR56 by flow cytometry. LAG-3, CXCR6, and CD69 frequencies were enriched on synovial GPR56+ CD4+ T cells (Fig. 4b, c, gating strategy in Supplementary Fig. 1c) in ACPA+ SF. In contrast, CD49a frequency was higher on GPR56− CD4+ T cells. In ACPA− SF, GPR56+ CD4+ T cells also showed a higher frequency of CXCR6, CD69, and LAG-3 as compared to GPR56− CD4+ T cells. However, LAG-3 frequency was higher on CD4+

**Fig. 3 GPR56 expression on peripheral helper CD4$^+$ T cells in ACPA+ and ACPA− RA SF. a** 2-D dot plots showing the expression of selected genes in the different CD4$^+$ T-cell clusters in ACPA+ and ACPA− SF (circle size indicates the percentage of cells expressing, color intensity indicates average expression) ($n = 4$ ACPA+, $n = 4$ ACPA− RA SF). **b** Representative flow cytometry dot plot of GPR56 expression within PD-1$^{high}$ (T$_{PH}$) and PD-1$^−$ (non-T$_{PH}$) CD4$^+$ T cells in ACPA+ (upper panel) and ACPA− (lower panel) RA SF, quantified in $n = 9$ ACPA− SF and $n = 12$ ACPA+ SF, $P = 0.0006$ (% PD-1$^{high}$), $P < 0.0001$ (%GPR56 in PD-1$^−$ versus PD-1$^{high}$ in ACPA+), $P < 0.0001$ (%GPR56 in PD-1$^−$ versus PD-1$^{high}$ in ACPA−), $P = 0.0007$ (%GPR56 in PD-1$^{high}$ in ACPA− versus ACPA+). **c** Left panel, representative flow cytometry dot plot of the two T$_{PH}$ states (PD-1$^{high}$GPR56$^{low}$ and PD-1$^{high}$GPR56$^+$) and quantification of their frequency in SF, ns (%PD-1$^{high}$GPR56$^{low}$), $P = 0.0005$ (%PD-1$^{high}$GPR56$^+$) in ACPA+ versus ACPA−. Middle panel, MHC-II MFI: $P < 0.0001$ (PD-1$^−$GPR56$^−$ versus PD-1$^{high}$GPR56$^{low}$), $P < 0.0001$ (PD-1$^−$GPR56$^−$ versus PD-1$^{high}$GPR56$^+$) in ACPA− SF; $P = 0.0001$ (PD-1$^−$GPR56$^−$ versus PD-1$^{high}$GPR56$^{low}$), $P < 0.0001$ (PD-1$^−$GPR56$^−$ versus PD-1$^{high}$GPR56$^+$), $P = 0.0192$ (PD-1$^{high}$GPR56$^{low}$ versus PD-1$^{high}$GPR56$^+$) in ACPA+ SF; $P = 0.0381$ (PD-1$^{high}$GPR56$^{low}$ in ACPA− versus ACPA+), ns (PD-1$^{high}$GPR56$^+$ in ACPA− versus ACPA+). Right panel, PD-1 MFI: $P < 0.0001$ (PD-1$^−$GPR56$^−$ versus PD-1$^{high}$GPR56$^{low}$), $P < 0.0001$ (PD-1$^−$GPR56$^−$ versus PD-1$^{high}$GPR56$^+$) in ACPA− SF; $P < 0.0001$ (PD-1$^−$GPR56$^−$ versus PD-1$^{high}$GPR56$^{low}$), $P < 0.0001$ (PD-1$^−$GPR56$^−$ versus PD-1$^{high}$GPR56$^+$), $P = 0.0059$ (PD-1$^{high}$GPR56$^{low}$ versus PD-1$^{high}$GPR56$^+$) in ACPA+ SF; $P = 0.0491$ (PD-1$^{high}$GPR56$^{low}$ in ACPA− versus ACPA+), $P = 0.0056$ (PD-1$^{high}$GPR56$^+$ in ACPA− versus ACPA+). ($n = 9$ ACPA−, $n = 11$ ACPA+, MHC-II) ($n = 9$ ACPA−, $n = 12$ ACPA+, PD-1). **d** Representative flow cytometry dot plots of CXCL13 and BLIMP-1 expression within PD-1$^−$GPR56$^−$ (non-T$_{PH}$) and PD-1$^{high}$GPR56$^{low}$ and PD-1$^{high}$GPR56$^+$ (2 T$_{PH}$ states) in CD4$^+$ T cells in ACPA+ and ACPA− RA SF, quantified in **e** in $n = 5$ ACPA− SF and $n = 5$ ACPA+ SF. Upper panel, %CXCL13: $P = 0.0079$ (PD-1$^−$GPR56$^−$ versus PD-1$^{high}$GPR56$^{low}$), $P = 0.0079$ (PD-1$^−$GPR56$^−$ versus PD-1$^{high}$GPR56$^+$), $P = 0.0079$ (PD-1$^{high}$GPR56$^{low}$ versus PD-1$^{high}$GPR56$^+$) in ACPA− SF; $P = 0.0079$ (PD-1$^−$GPR56$^−$ versus PD-1$^{high}$GPR56$^{low}$), $P = 0.0079$ (PD-1$^−$GPR56$^−$ versus PD-1$^{high}$GPR56$^+$), $P = 0.0317$ (PD-1$^{high}$GPR56$^{low}$ versus PD-1$^{high}$GPR56$^+$) in ACPA+ SF; ns (PD-1$^{high}$GPR56$^+$ in ACPA− versus ACPA+). Lower panel, % BLIMP-1: $P = 0.0159$ (PD-1$^−$GPR56$^−$ versus PD-1$^{high}$GPR56$^{low}$), $P = 0.0079$ (PD-1$^−$GPR56$^−$ versus PD-1$^{high}$GPR56$^+$), ns (PD-1$^{high}$GPR56$^{low}$ versus PD-1$^{high}$GPR56$^+$) in ACPA− SF; $P = 0.0079$ (PD-1$^−$GPR56$^−$ versus PD-1$^{high}$GPR56$^{low}$), $P = 0.0079$ (PD-1$^−$GPR56$^−$ versus PD-1$^{high}$GPR56$^+$), $P = 0.0079$ (PD-1$^{high}$GPR56$^{low}$ versus PD-1$^{high}$GPR56$^+$) in ACPA+ SF; ns (PD-1$^{high}$GPR56$^+$ in ACPA− versus ACPA+). **b**, **c** Data are from a pool of nine independent experiments where a circle is a single replicate. **e** Data are from a pool of five independent experiments where a circle is a single replicate. **b**, **c**, **e** Line represents median, two-tailed Mann–Whitney $U$ test. Blue dots indicate ACPA− RA SF and red dots indicate ACPA+ RA SF. ns not significant, MFI mean fluorescence intensity.

T cells in ACPA+ SF as compared to ACPA− SF (Fig. 4d) and correlated with CCP positivity (Fig. 4e). In CD8$^+$ T cells, CD49a was the only receptor for which an increased frequency was observed in ACPA+ SF (Supplementary Fig. 14). These data indicate that, in SF, GPR56 expression correlates with the concomitant expression of a distinct set of tissue-resident memory receptors on CD4$^+$ T cells: CXCR6 and CD69 and the inhibitory receptor LAG-3.

**Clonally expanded CXCL13$^{high}$ T$_{PH}$ CD4$^+$ T cells in RA synovial fluid.** Next, we examined the expanded TCR clones in relation to the aforementioned CD4$^+$ T-cell clusters (Supplementary Data 4 and 5). A clone was defined by at least 2 cells sharing the same CDR3 amino acid sequence from both TCRα and TCRβ chains (Supplementary Data 6a). Of note, TCR recovery was low in patient RA12 which was excluded from further clonality analysis. In SF, the most expanded clones were found in CXCL13$^{high}$ T$_{PH}$ (median frequency 45.7%), Treg (median frequency 28.7%), effector CD4$^+$ (median frequency 11.3%) and cytotoxic CD4$^+$ (median frequency 7.2%) T-cell clusters (Fig. 5a–c and Supplementary Data 6b). In all four ACPA+ RA patients CXCL13$^{high}$ T$_{PH}$ cells contained most of the expanded clones. In two out of three ACPA− RA patients (RA8, RA9), Tregs represented the most expanded clones in SF (Fig. 5b, left panel, Fig. 5c, left panel and Supplementary Data 6b). This observation prompted us to analyze the frequency of FOXP3$^+$ CD4$^+$ T cells which was slightly increased in ACPA− SF (Fig. 5d). FOXP3$^+$ CD4$^+$ T-cell frequency in SF was not different depending on treatment at the time of sampling (Supplementary Fig. 7b). In blood, most of the expanded clones originated from central memory CD4$^+$ T cells (median frequency 45.8%) and cytotoxic CD4$^+$ T cells (median frequency 32.6%) (Fig. 5b, right panel, Fig. 5c, right and Supplementary Data 6c) irrespective of their ACPA status.

Overall, these data show that in SF of RA patients most of the expanded clones are found within the CXCL13$^{high}$ T$_{PH}$, the Tregs, the effector and the cytotoxic CD4$^+$ T-cell subsets. In blood, the expanded T-cell clones were mostly found within cytotoxic and central memory CD4$^+$ T cells.

**The two T$_{PH}$ subsets are functionally related.** Further, we examined the overlapping of all CDR3 sequences between SF CD4$^+$ T-cell subsets that would be indicative of a possible shared differentiation pathway (Fig. 6a, b). Of note, no common CDR3 sequences were observed across patients. In SF, CXCL13$^{high}$ T$_{PH}$ showed clonal overlap with CXCL13$^{low}$ T$_{PH}$ (Jaccard Index (JI):0.038), proliferating (JI:0.031), effector (JI:0.023) and cytotoxic CD4$^+$ T cells (JI:0.013) (Fig. 6b). SF cytotoxic CD4$^+$ T cells showed also clonal overlap with effector (JI:0.016) and proliferating CD4$^+$ T cells (JI:0.013). CXCL13$^{low}$ T$_{PH}$ presented a clonal overlap with CXCL13$^{high}$ T$_{PH}$ (JI:0.038), proliferating (JI:0.028), effector (JI:0.014), and cytotoxic CD4$^+$ T cells (JI:0.007). SF Tregs showed clonal overlap with proliferating CD4$^+$ T cells (JI:0.02). We then focused our analysis on expanded clones in CXCL13$^{high}$ T$_{PH}$, Tregs, effector CD4$^+$ T cells and cytotoxic CD4$^+$ T cells (Supplementary Data 7). A median of 59% of the CXCL13$^{high}$ T$_{PH}$ expanded clones shared CDR3 sequences with cells from other clusters (either clones or unique cells) (Supplementary Data 6d). These shared sequences were identified in all patients. A median frequency of 28% of these clones was shared with proliferating CD4$^+$ T cells, 25% with CXCL13$^{low}$ T$_{PH}$, 19% with effector CD4$^+$ T cells, and 6% with cytotoxic CD4$^+$ T cells (Fig. 6c and Supplementary Data 6e). Treg clones shared a median of 42% and 17% of their CDR3 sequences with proliferating CD4$^+$ T cells and CXCL13$^{high}$ T$_{PH}$ cells, respectively. Effector CD4$^+$ T-cell clones shared a median of 26% and 10% of their CDR3 sequences with CXCL13$^{high}$ T$_{PH}$ and cytotoxic CD4$^+$ cells, respectively. Cytotoxic CD4$^+$ T-cell clones shared a median of 26% and 6% of their CDR3 sequences with CXCL13$^{high}$ T$_{PH}$ and effector CD4$^+$ T cells. Of note, for effector and cytotoxic CD4$^+$ T cells, sharing was primarily coming from two patients (RA10, and RA11, Supplementary Data 6e and 7). Cytotoxic CD4$^+$ T cells from PB and SF also presented clonal overlap (JI:0.036). To further understand the cell fate of CXCL13$^{low}$ T$_{PH}$ cells in the context of cytotoxic CD4$^+$ T cells, we sorted GPR56$^{low}$ PD-1$^{high}$ CD4$^+$ T$_{PH}$ cells and stimulated with CD3/CD28 beads (gating strategy in Supplementary Fig. 1D). We observed an increased PD-1 expression and a tendency towards increased GPR56 and BLIMP-1 expression (Fig. 6d). Perforin-1

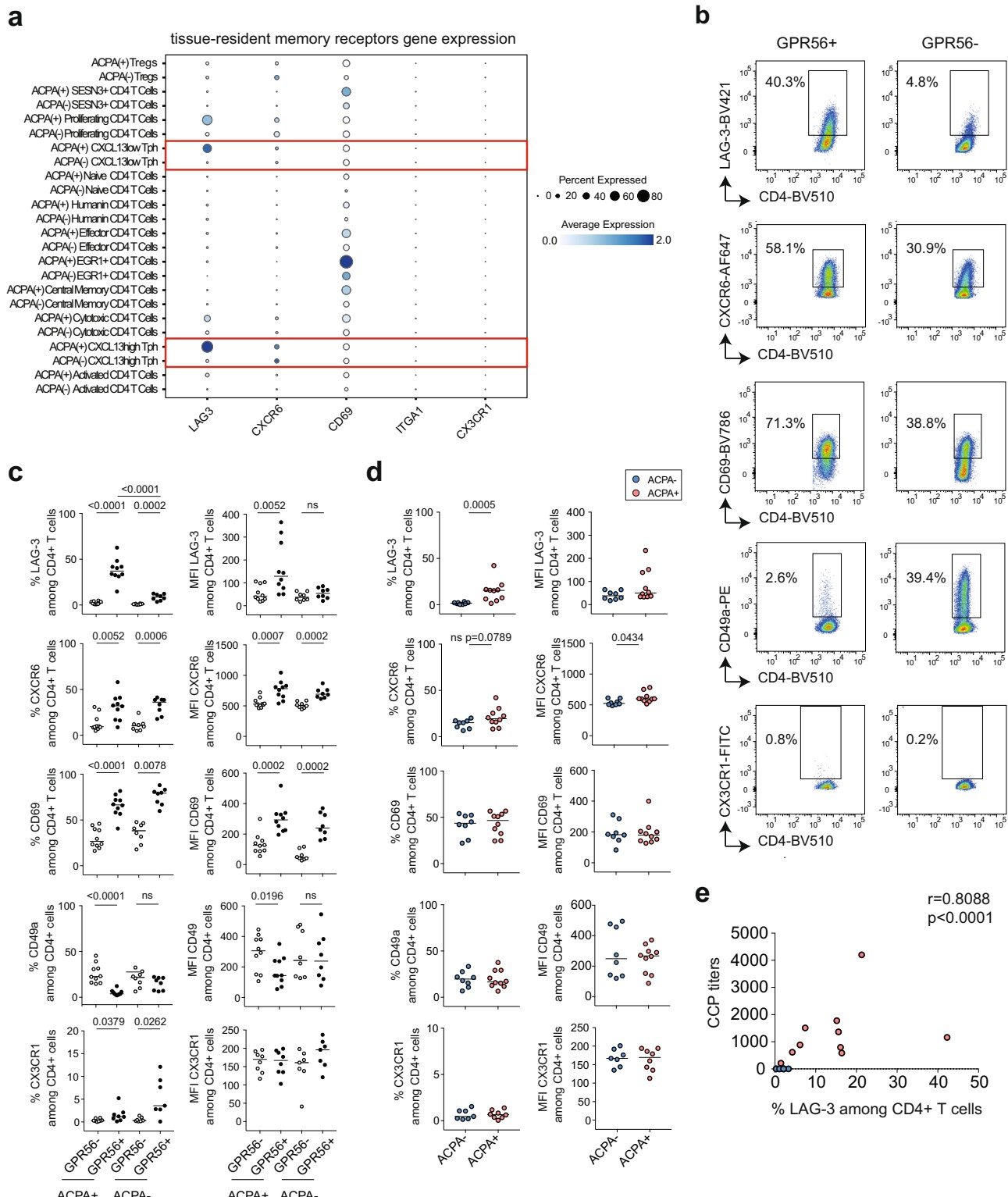

expression, however, did not dramatically increase. Of note, CXCL13 did not increase after TCR stimulation, possibly due to the lack of additional inducers such as TGFβ and IL-2 neutralizing conditions as previously described[26]. To further evaluate cell differentiation across clusters, we performed an RNA-velocity analysis (scVelo) which estimates future cell states based on the ratio of detected spliced and unspliced mRNAs transcripts[33]. Unsurprisingly, we observed a progression from naive CD4+

T cells into central memory and effector CD4+ T cells validating our approach (Fig. 6e). In ACPA+ SF, both CXCL13low T$_{PH}$ and effector CD4+ T cells led to CXCL13high T$_{PH}$ cells (Fig. 6e, right panel). This transition from CXCL13low to CXCL13high T$_{PH}$ was less obvious in ACPA− SF (Fig. 6e, left panel). In addition, the cytotoxic CD4+ T-cell final differentiation stage was originating from SESN3+ and effector CD4+ T cells in both ACPA− and ACPA+ RA. Hence, effector CD4+ T cells project into two

**Fig. 4 Tissue-resident memory receptors on CD4+ T cells in ACPA+ and ACPA− RA SF. a** 2-D dot plots showing the expression of selected genes in the different CD4+ T-cell clusters in ACPA+ and ACPA− SF (circle size indicates the percentage of cells expressing, color intensity indicates average expression) (n = 4 ACPA+, n = 4 ACPA− RA SF). **b** Representative flow cytometry dot plots showing the expression of LAG-3, CXCR6, CD69, CD49a, and CX3CR1 on GPR56+ and GPR56− CD4+ T cells in ACPA+ RA, quantified in ACPA+ and ACPA− SF in **c**) P < 0.0001 (%LAG-3), P = 0.0052 (% CXCR6), P < 0.0001 (%CD69), P < 0.0001 (%CD49a), P = 0.0379 (%CX3CR1) in GPR56− versus GPR56+ in ACPA+ SF; P = 0.0002 (%LAG-3), P = 0.0006 (%CXCR6), P = 0.0078 (%CD69), ns (%CD49a), P = 0.0262 (%CX3CR1) in GPR56− versus GPR56+ in ACPA− SF; P < 0.0001 (%LAG-3) in GPR56+ in ACPA− versus ACPA+; P = 0.0052 (MFI LAG-3), P = 0.0007 (MFI CXCR6), P = 0.0002 (MFI CD69), P = 0.0196 (MFI CD49a) in GPR56− versus GPR56+ in ACPA+ SF; ns (MFI LAG-3), P = 0.0002 (MFI CXCR6), P = 0.0002 (MFI CD69), ns (MFI CD49a) in GPR56− versus GPR56+ in ACPA − SF; (n = 10 ACPA+, n = 8 ACPA−, LAG-3, CXCR6, CD69, CD49a) (n = 8 ACPA+, n = 7 ACPA−, CX3CR1). White dots indicate GPR56− CD4+ T cells and black dots indicate GPR56+ CD4+ T cells. **d** Expression of LAG-3, CXCR6, CD69, CD49a and CX3CR1 on CD4+ T cells in SF; P = 0.0005 (% LAG-3 in ACPA− versus ACPA+); (n = 8 ACPA−, n = 10 ACPA+, LAG-3, CXCR6, CD69, CD49a) and (n = 7 ACPA−, n = 8 ACPA+, CXC3CR1). Blue dots indicate ACPA− RA SF and red dots indicate ACPA+ RA SF. **c, d** Line represents median, two-tailed Mann–Whitney U test. Data are from a pool of eight independent experiments where a circle is a single replicate. **e** Correlation between the frequency of LAG-3 in CD4+ T cells in RA SF and the levels of anti-CCP (cyclic citrullinated peptide) antibodies (n = 18), P < 0.0001, Spearman two-tailed test. ns not significant.

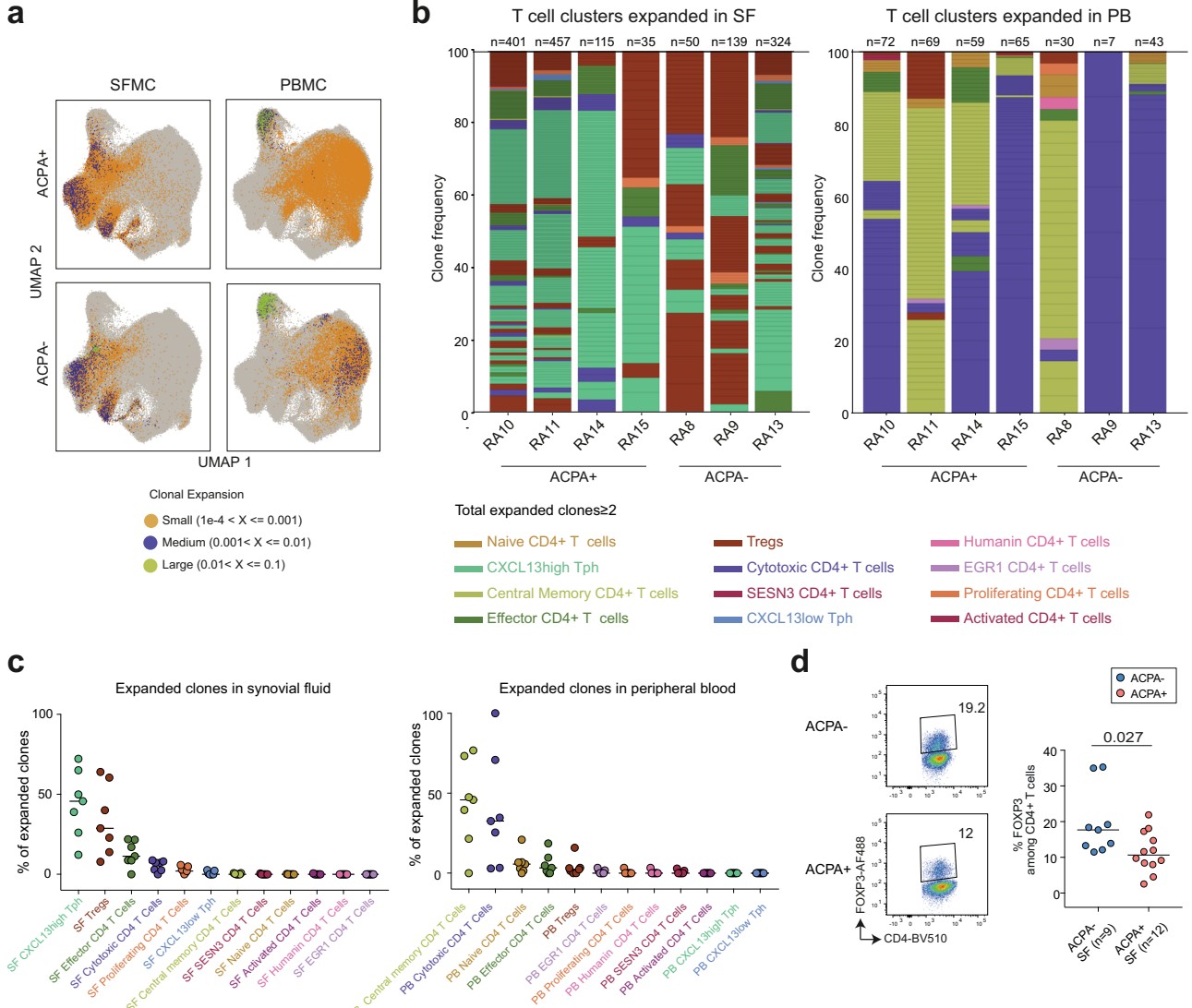

**Fig. 5 Clonally expanded CD4+ T cells in RA. a** UMAP plots showing expanded clones in ACPA+ (n = 4) and ACPA− (n = 4) RA PB and SF, colored based on sample size. Orange dots indicate small clones (1e-4 < X ≤0.001), purple dots indicate medium size clones (0.001 < X ≤0.01), green dots indicate large size clones (0.01 < X ≤0.1). Each dot displays a cell. **b** Stacked barplots displaying the phenotype of the SF (left) and PB (right) expanded clones (n ≥ 2 cells) in each RA patient, quadrant represents an individual clone, total number of clones per patient are indicated above each column. **c** Frequency of expanded clones within each CD4+ T-cell cluster in SF (left panel) and PB (right panel) for each RA patient (n = 4 ACPA+, 3 ACPA−). **d** Frequency of FOXP3+ cells among CD4+ T cells in ACPA− (n = 9) and ACPA+ (n = 12) RA SF, P = 0.027. Data are from a pool of nine independent experiments where a circle is a single replicate. Blue dots indicate ACPA− RA SF and red dots indicate ACPA+ RA SF. Line represents median, two-tailed Mann–Whitney U test.

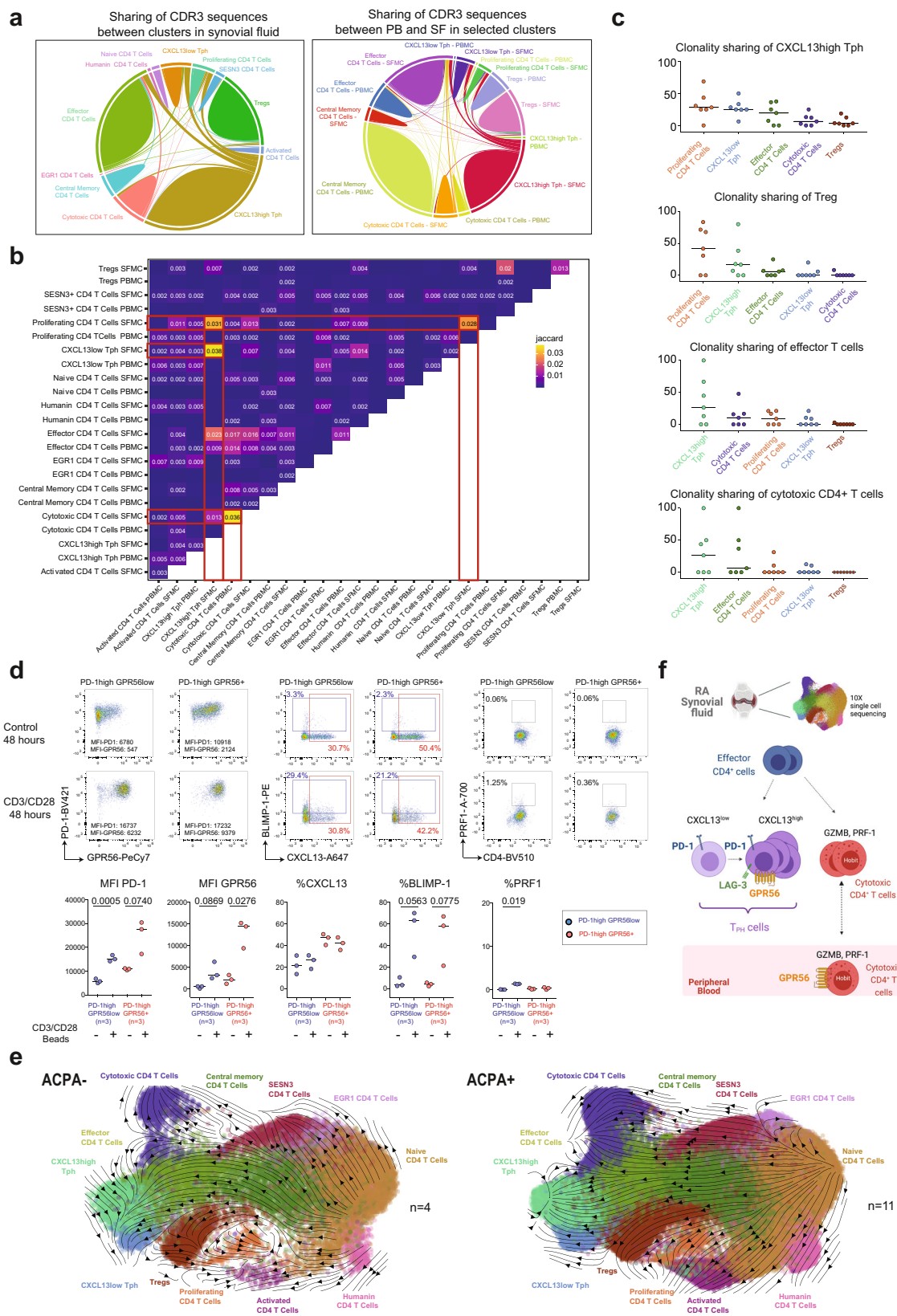

independent branches leading to $T_{PH}$ in one direction and cytotoxic CD4$^+$ T cells in the other direction. Tregs were connected to central memory and proliferative CD4$^+$ T cells. These lines of evidence suggest that CXCL13$^{low}$ and CXCL13$^{high}$ $T_{PH}$ in SF are functionally and developmentally related, possibly originating from effector CD4$^+$ T cells while cytotoxic CD4$^+$ T cells potentially arise from other effector CD4$^+$ T cells independent of the $T_{PH}$ state.

**Fig. 6 CD4+ T-cell subsets interconnectivity. a** Chord diagrams showing the interconnectivity between CD4+ T cells sharing the same CDR3 (paired TCRα, and TCRβ chains) amino acid sequences within CD4+ T-cell clusters in RA SF (left panel) and between compartments in selected clusters (right panel) ($n = 4$ ACPA−, $n = 4$ ACPA+). **b** Jaccard overlap quantifications for clonal overlap between cell clusters across compartments ($n = 4$ ACPA−, $n = 4$ ACPA+). **c** Percentage of shared expanded clones ($n \geq 2$ cells) shared with selected subsets among total cell clones, $n = 7$ RA patients ($n = 4$ ACPA+, 3 ACPA−). **d** Representative flow cytometry dot plots showing the expression of PD-1, GPR56, CXCL13, BLIMP-1, and PRF1 in control (upper panel) or after 48 h CD3/CD28-beads activation of $T_{PH}$ cell states (lower panel) (PD-1$^{high}$GPR56$^{low}$ and PD-1$^{high}$GPR56$^+$ from ACPA+ SF), quantified in $n = 3$ ACPA+ RA patients. **d** Data are from a pool of three independent experiments where a circle is a single replicate. Line indicates median, two-tailed Wilcoxon paired test, $P = 0.0005$ (MFI PD-1 on PD-1$^{high}$GPR56$^{low}$ after activation), $P = 0.0740$ (MFI PD-1 on PD-1$^{high}$GPR56$^+$ after activation), $P = 0.0869$ (MFI GPR56 on PD-1$^{high}$GPR56$^{low}$ after activation), $P = 0.0276$ (MFI GPR56 on PD-1$^{high}$GPR56$^+$ after activation), $P = 0.0563$ (% BLIMP-1 on PD-1$^{high}$GPR56$^{low}$ after activation), $P = 0.0775$ (% BLIMP-1 on PD-1$^{high}$GPR56$^+$ after activation), $P = 0.019$ (% PRF1 on PD-1$^{high}$GPR56$^{low}$ after activation). **e** Velocity plots showing the phenotype directionality between the different CD4+ T-cell clusters in RA split in ACPA− (left panel) and ACPA+ (right panel) ($n = 4$ ACPA−, $n = 11$ ACPA+, pool PB and SF). **f** Graphical summary showing the proposed developmental link between the two $T_{PH}$ subsets, effector and cytotoxic CD4+ T cells in SF as well as the identified receptors.

## Discussion

Leveraging single-cell technology of CD4+ T cells in PB and SF of RA patients, we dissected the $T_{PH}$ CD4+ T-cell subset at the single-cell level showing that it is composed of two independent clusters with different *CXCL13* expression levels. So far, the $T_{PH}$ CD4+ T-cell subset was only defined by high expression of PD-1 and MHC-II while more accurate markers were still lacking. We show that GPR56 clearly delineates the CXCL13$^{high}$ $T_{PH}$ subset and associates with the expression of tissue-resident memory receptors CXCR6 and CD69 and the inhibitory receptor LAG-3. We also identify an increased frequency of cytotoxic CD4+ T cells in ACPA+ RA SF. Finally, we show that CXCL13$^{high}$ $T_{PH}$, Treg, T effector, and cytotoxic CD4+ T cells are clonally expanded in RA SF. Together, these data provide a refined characterization of $T_{PH}$ and cytotoxic CD4+ T cells, two important CD4+ T-cell subsets in RA (Fig. 6f).

In synovial tissue of ACPA+ RA patients, $T_{PH}$ CD4+ cells produce CXCL13[7,9,34], localize with B cells and have the ability to induce in vitro plasma cell differentiation through IL-21 production[7]. $T_{PH}$ cells are defined by high expression of PD-1 and MHC-II, the cytokines CXCL13, IL-21, and IL-10, and the transcription factors BLIMP-1, TOX, MAF, and SOX4[7,9,26]. $T_{PH}$ CD4+ T-cell differentiation is induced by TGFβ in IL-2 neutralizing conditions[26]. Single-cell sequencing identified two distinct clusters of $T_{PH}$ cells: CXCL13$^{low}$ $T_{PH}$ cells and CXCL13$^{high}$ $T_{PH}$ cells. We postulated that these two clusters correspond to two states of differentiation: CXCL13$^{low}$ $T_{PH}$ CD4+ T cells could be the precursors of the more differentiated cytokine-producing CXCL13$^{high}$ $T_{PH}$ CD4+ T cells. In line with this hypothesis, common TCR sequences were observed between CXCL13$^{low}$ $T_{PH}$ cells and CXCL13$^{high}$ $T_{PH}$ cells clusters suggesting a common origin between these two subsets. CD3/CD28 stimulation of GPR56$^{low}$ PD-1$^{high}$ $T_{PH}$ cells also induced an increased PD-1 upregulation compatible with this scenario. Finally, RNA velocity and TCR clonality analysis show a connection between effector memory CD4+ T cells and CXCL13$^{high}$ $T_{PH}$, indicating that the precursors of $T_{PH}$ cells are probably contained within the effector CD4+ T-cell population which is enriched in synovial fluid.

CXCL13$^{high}$ $T_{PH}$ CD4+ T cells were also detected in ACPA− RA SF but at a much lower frequency than in ACPA+ patients. In ACPA− patients, these few CXCL13$^{high}$ $T_{PH}$ also presented an increased expression of bona fide $T_{PH}$ markers (MHC-II, PD-1, CXCL13, BLIMP-1) but lower levels of LAG-3, PD-1, and GPR56 than in ACPA+ patients. The expression of these inhibitory receptors might be a sign of persistent T-cell activation in the ACPA+ synovial joint. We also identified clonally expanded Tregs in SF of two ACPA− RA patients and a slight increase in Treg frequency. Altogether, these data suggest that the differentiation towards CXCL13$^{high}$ $T_{PH}$ cells is increased in the joint of ACPA+ RA as compared to ACPA− RA patients. In line with this hypothesis, the RNA-velocity analysis of ACPA+ CD4+ T cells revealed a trajectory from CXCL13$^{low}$ to CXCL13$^{high}$ $T_{PH}$. An increased frequency of PD-1$^{high}$ CD4+ T cells was recently identified in leukocyte-rich RA synovial tissues[8]. We confirm the presence of this subset also in SF with an increased frequency in ACPA+ RA patients. While SF analysis allows a refined analysis of many immune cells and the downstream clonality analysis, a limitation is that these samples are originating from patients with long disease duration and under immunosuppressive drugs. In our dataset, we did not find any specific association with ongoing medication but, interestingly, the only ACPA− patient whose SF presented CXCL13$^{high}$ $T_{PH}$ cell clonal expansion was not under any immunosuppressive medication at time of sampling. Longitudinal studies will inform about the possible effects of current treatments on the expansion and maintenance of these CD4+ T-cell subsets.

Expanded TCR clones have been identified in synovial tissues and fluid from RA patients using bulk TCRβ sequencing[5]. In blood, expanded cytotoxic CD4+ CD8$^{null}$ cells present with a bias in their TCRVβ usage[35,36]. We found that expanded clones in the blood were originating from the CD4+ cytotoxic T-cell subset. Clonally expanded cytotoxic CD4+ T cells were also observed in RA SF, although at a lower degree than the CXCL13$^{high}$ $T_{PH}$ cells. Overlap clonality between cytotoxic clones in PB and SF suggest their recirculation. Expansions of cytotoxic CD4+ T cells are frequently observed in chronic viral infections, and they are capable of directly killing EBV-infected B cells in an MHC-II-dependent manner[37]. Our data show that cytotoxic CD4+ T cells are enriched in SF of ACPA+ RA patients and correlate with CCP titers, suggesting that these cells are mainly implicated in the subset of autoantibody-positive RA. Still, so far, their autoreactive properties have not been investigated in the context of citrulline immunity. The differentiation processes that lead to the differentiation of cytotoxic CD4+ T cells are not completely elucidated, but IL-2[38], IL-15[39], and type I IFNs[40] contribute to their development. The RNA-velocity analysis suggests that cytotoxic CD4+ T cells originate from effector CD4+ T cells and from the subset of SESN3$^+$ memory T cells. Although the memory SESN3$^+$ CD4+ T-cell subset has not been extensively studied before, it has been identified in skin biopsies from patients with psoriasis[41]. Interestingly, we detected cytokine-related signatures in SESN3$^+$ CD4+ T cells which could be implicated in their differentiation into cytotoxic CD4+ T cells. Further studies are warranted to dissect the differential pathways leading to cytotoxic CD4+ and CXCL13$^{high}$ $T_{PH}$ cells, two CD4+ T-cell types enriched in ACPA+ RA.

GPR56 is an adhesion G-protein-coupled receptor encoded by the *ADGRG1* gene implicated in numerous migration/adhesion processes including neuron migration[42] and tumor growth inhibition[43]. In NK cells, GPR56 is an inhibitory receptor binding

to the tetraspanin CD81[29]. We confirmed data from a previous report showing that GPR56 is also a marker of peripheral cytotoxic CD4+ cells[18]. At the site of inflammation, however, we discovered that GPR56 was highly expressed on T$_{PH}$ cells in ACPA+ RA patients and could be used to identify T$_{PH}$ CD4+ T cells. In SF CD4+ T cells, GPR56 expression correlated with the expression of the inhibitory receptors LAG-3 and PD-1 as well as CXCR6 and CD69, receptors expressed on tissue-resident memory T cells[30]. Similarly, in multiple sclerosis, CD8+ T cells from active lesions exhibit a tissue-resident memory phenotype (CD69, CD103, PD-1, CD49a) associated with an intermediate expression of GPR56 without granzyme B expression[44]. In mice, Blimp-1 and Hobit mediates the transcriptional program associated with T-cell tissue residency[45]. In ACPA+ SF, BLIMP-1 was expressed on T$_{PH}$ CD4+ cells presenting with tissue-resident memory receptors whereas Hobit was only expressed on cytotoxic CD4+ T cells suggesting that, BLIMP-1 could drive tissue residency of T$_{PH}$ CD4+ T cells in the synovial joint. GPR56 might be implicated in the migration and/or maintenance of T$_{PH}$ CD4+ T cells in the synovial joints where its ligand remains to be determined. In SF, cytotoxic CD4+ T cells expressed low levels of *ADGRG1* and had no detectable GPR56 surface expression. We showed that similar to what has been described in NK cells[29], PMA/Ionomycin stimulation induced a rapid GPR56 downregulation on GPR56+ CD4+ T cells. In vivo, the inflammatory milieu might therefore contribute to GPR56 downregulation on SF cytotoxic CD4+ T cells.

In summary, we identify two clusters of T$_{PH}$ CD4+ T cells and observe clonally expanded CXCL13$^{high}$ T$_{PH}$, Tregs, effector, and cytotoxic CD4+ T cells in RA SF. We show that GPR56 is a receptor that should be used to identify CXCL13$^{high}$ T$_{PH}$ CD4+ cells. Future studies aiming at understanding the function of GPR56 in T$_{PH}$ CD4+ and cytotoxic CD4+ T cells will contribute to evaluating the feasibility of targeting this receptor in ACPA+ RA.

## Methods

**Patient and samples**. This study was approved by the research ethics committee of Karolinska University Hospital, Sweden and all patients signed informed consent according to the Declaration of Helsinki. All patients had established disease, based on the 1987 criteria from the American College of Rheumatology. Synovial fluid from ACPA− (n = 13) and ACPA+ (n = 16) RA patients were collected at several visits at the Rheumatology Clinic of Karolinska University Hospital, Stockholm (Supplementary Data 1). In n = 12 ACPA− and n = 10 ACPA+ RA patients, paired PBMCs were also available. PBMC and SFMC were isolated by Ficoll separation (GE Healthcare) and cryopreserved in fetal bovine serum (FBS) with 10% DMSO in liquid nitrogen.

**Flow cytometry**. In total, $1 \times 10^6$ PBMC and $1 \times 10^6$ SFMC were labeled with LIVE DEAD Fixable NEAR IR Dead Cell Stain Kit (Invitrogen, ref: L10119) and further stained with fluorescently labeled antibodies (Supplementary Data 8). ACPA− and ACPA+ SF and PB were always run in the same experiment. Depending on the number of recovered cells, 2–5 flow cytometry panels were run (Supplementary Data 9). For intracellular stainings, cells were fixed and permeabilized using the FOXP3 permeabilization buffer kit (ebioscience, ref: 00–5523) according to the manufacturer´s protocol. In some conditions, $1 \times 10^6$ PBMC and $1 \times 10^6$ SFMC were activated with CD3/CD28 dynabeads (ThermoFischer, ref: 11131D, one bead/cell) or Phorbol-12-myristate-13-acetate (PMA, Calbiochem, ref: 10,524400 ng/mL) plus Ionomycin (Calbiochem, ref: 1,407950 μM) in RPMI 10% FBS in the presence of Brefeldin-A (Sigma-Aldrich, ref: B6542, 5 μg/mL) for 3 h. Samples were run on the BD LSR Fortessa and flow cytometry data were analyzed with the FlowJo software (version 10.07.1) (gating strategies, Supplementary Fig.1).

**Sorting and T-cell activation**. SFMC (n = 3, ACPA+) were labeled with LIVE DEAD Fixable NEAR IR Dead Cell Stain Kit (Invitrogen, ref: L10119) and further stained with fluorescently labeled antibodies (Supplementary Data 8, Supplementary Data 9, panel 6, Supplementary Fig. 1). GPR56$^{low}$PD-1$^{high}$ and GPR56+PD-1$^{high}$ T$_{PH}$ were subsequently cell sorted on the BD Influx. Sorted cells were incubated with CD3/CD28 dynabeads (1bead/cell, ThermoFischer, ref: 11131D) in RPMI 10%FBS for 48 h. Brefeldin-A (Sigma-Aldrich, ref: B6542, 5 μg/mL) was added during the last 4 h. Samples were then processed as described in "Flow cytometry".

**Statistical analysis**. Flow cytometry results were analyzed with the Prism 7 software (GraphPad, San Diego, CA, USA). Statistical tests performed are indicated in the figure legends.

**Single-cell RNA sequencing of RA samples**. CD4+ T cells were enriched from cryopreserved PBMC and SFMC samples using the EasySep human CD4 Positive Selection Kit II (STEMCELL Technologies, Catalog# 17852) (Supplementary Fig. 3). In experiment 10X-1 (Supplementary Data 1), for samples from RA1–7, following CD4+ T-cell isolation, cells were counted, hashed with TotalSeq-C0251 (ref: 394661) and TotalSeq-C0253 (ref: 394665) anti-human antibodies, Biolegend) and pooled per patient (i.e., pool 1 = PB and SF from patient 1, pool 2 = PB and SF from patient 2, etc.). The pooled cells were then counted again and immediately loaded onto the chromium chip G using the standard protocol for the Chromium single-cell 5' kit v1 (10x Genomics, Inc). Following Gel Bead-in Emulsion (GEM) generation, samples were processed using the standard manufacturer's protocol. This pipeline was also used to process samples for experiment 10X-2 (Supplementary Data 1) with RA8–15 except for antibody hashing and the chromium kit version; each sample was processed independently (i.e., 1 sample/10x chromium lane; Supplementary Fig. 3) and all cells were processed with the Chromium single-cell 5' kit v2 (Dual Index). Once sequencing libraries passed standard quality control metrics, the libraries for RA1–7 were sequenced on Illumina NextSeq500/550 high output 75-cycle v2.5 kits with the following read structure: read1: 26, read2: 58, index 1: 8. Libraries for RA8–15 were sequenced on Illumina NextSeq500/550 high output 150-cycle v2.5 kits with the following read structure: read1: 26, read2: 90, index 1: 10, index 2: 10. Both sets of libraries were sequenced to obtain a read depth greater than 20,000 reads/cell for the gene-expression (GEX) libraries and greater than 5000 reads/cell for the V(D)J-enriched T Cell libraries.

**Genomic alignment and digital-gene-expression generation**. Following sequencing, the pooled GEX libraries were demultiplexed using bc2fastq (v2.17) with parameters (–minimum-trimmed-read-length = 10–mask-short-adapter-reads = 10–ignore-missing-positions–ignore-missing-controls) to generate FASTQ files for each patient. FASTQ files were then uploaded to Terra (www.app.terr.bio) where the raw sequencing data were mapped and quantified using STAR within the 10X Genomics Inc software package CellRanger on Cumulus (https://cumulus.readthedocs.io/en/latest/cellranger.html, snapshot 15, Cellranger 5.0.1, default parameters) with GRCh38-2020-A (https://cumulus.readthedocs.io/en/latest/cellranger.html) to generate UMI-collapsed, gene-by-cell expression matrices. For the RNA-velocity analysis, exonic and intronic alignment information was retrieved using velocyto on SevenBridges (https://www.sevenbridges.com/) and results visualized using scVelo[33,46].

**Data preprocessing and quality control**. Aligned matrices were first filtered to remove low-quality barcodes, keeping only those with greater than 200 UMIs and less than 25% mitochondrial reads. After removing low-quality barcodes, cell doublets were then computationally removed using DoubletFinder[47] with calculated doublet rates based on cell input for each sample. Using cell-type-defining markers, the cleaned datasets were filtered for non-CD4 T cells (e.g., monocytes, macrophages, CD8+ T, CD20+ B cells using *CD14, CD8a, CD8b*, and *MS4A1*). Finally, we attempted to deconvolute the multiplexed PBMC and SFMC cell populations, from RA1–7, using antibody hashes; however, due to sub-optimal tagging, we were unable to separate the compartment cell populations. To prevent this complication from occurring with RA8–15, we omitted antibody hashing from our pipeline and processed compartments separately to retain compartment-specific cell populations. Samples from RA1–15 were then merged accordingly for the subsequent unsupervised (Fig. 2b, left panel) and velocity analyses (Fig. 6e and Supplementary Fig. 3).

**Unsupervised analysis and cell-type annotation**. Using the Seurat[19] (v4.0.0) package, filtered gene-by-cell matrices for RA1–15 were merged and then processed using a standard unsupervised workflow (i.e., normalization, scaling, dimensionality reduction, batch correction, cell clustering, and differential gene-expression analysis). First, the merged dataset was normalized and log-transformed (scaling factor = 10,000). The top 3000 highly variable genes (HVGs) were identified and used for scaling the data. During scaling, three sources of unwanted variation, cumulative UMI capture, cell cycling programs, and percent mitochondrial reads, were regressed out. Dimensionality reduction was then performed with principal component analysis (PCA) over the top 3000 variable genes and, using the JackStraw function in Seurat to identify statistically significant principal components (PCs, P < 0.05), as well as exploring individual PCs, we selected the top 70 principal components for visualization. Using these PCs, we corrected for patient-specific batch effects with the harmony[48] package, and then proceeded with constructing the nearest neighbor graph and Uniform Manifold Approximation and Projection (UMAP) plots. Cells were clustered using Louvain clustering and the package clustree[49] was used to generate a clustering tree and identify which resolution achieved stability. Differential gene-expression (DGE) analysis was then performed using the FindAllMarkers Seurat function with test.use set to 'MAST'[50]. Finally, to further remove non-T cells that passed our initial screen, clusters with (1) low expression of CD3, (2) differentially expressed genes (DEGs) that are

known markers of non-T cells, and (3) high enrichment scores for non-T-cell gene modules (MSigDB) were removed from the dataset; remaining cells were reanalyzed to reconstruct stable clusters for visualization and subsequent DGE analysis. Taking both the results of the DGE analysis and joint density approximations of known phenotypic markers, we identified 12 unique T-cell subsets: (1) naive CD4+ T cells (cluster 1; SELL+, CCR7+, IL7R+); (2) CXCL13high $T_{PH}$ (cluster 2; CXCL13+, ICOS+, CD3D+); (3) Central Memory CD4+ T cells (cluster 3; LTB+, KLF2+); (4) Effector CD4+ T cells (KLRB1+, GZMA+); (5) Tregs (cluster 5; FOXP3 + IL2Ra+); (6) Cytotoxic CD4+ T cells (cluster 6; NKG7+, GZMH+, PRF1+, and ZNF683+); (7) SESN3+ memory/effector CD4+ T cells (cluster 7; SESN3+); (8) CXCL13low $T_{PH}$ (cluster 8; CXCL13+, PRDM1+, CD3D+); (9) Humanin CD4+ T cells (cluster 9; MT-CO1+, MTRNR2L1+, MTRNR2L8+); (10) EGR1+ naïve CD4+ T cells (cluster 10; EGR1+); (11) Proliferating CD4+ T cells (cluster 11: MKI67+, STMN1+); (12) Activated CD4+ T cells (cluster 12; CD38+, HLA-DR+).

**RNA velocity**. The scVelo package, version 0.2.4[33,46] was used to model the RNA velocity of cells from volunteers RA1–15. Briefly, the RNA-velocity analysis leverages the expression relationship between spliced (exonic) and unspliced (intron-containing) RNA across selected variable genes to predict the directionality of transitions between cell types, as well as the putative driver genes of these transitions. Following the unsupervised analysis, the analyzed Seurat object was converted into an H5AD file using the SeuratDisk package [https://mojaveazure.github.io/seurat-disk/index.html], version 0.0.0.9011, and then imported into a jupyter notebook for preprocessing according to the scVelo standard pipeline: (1) genes were first normalized, selecting the top 3000 genes (pp.filter_-and_normalize(adata,n_top_genes = 3000, enforece = T), (2) principal components, nearest neighbors in PCA space, and first and second-order moments of the nearest neighbors were calculated (n_pcs = 70, n_neighbors = 20), (3) using the dynamical model from scVelo, RNA velocities were then estimated (tl.recover_dynamics(adata, mode = ' dynamical')) and (4) used to compute the velocities and velocity graph (tl.velocity and tl.velocity_graph). This approach was used for both ACPA+ and ACPA- RA cohorts.

**TCR mapping**. Following sequencing, V(D)J libraries for RA8–15 (experiment 10X-2, Supplementary Fig. 3, Supplementary Data 1) were demultiplexed using bc2fastq (v2.17.1.14) to generate FASTQ files for each sample, uploaded to terra, and mapped and quantified using the 10X Genomics Inc software package Cell-Ranger on Cumulus (https://cumulus.readthedocs.io/en/latest/cellranger.html, snapshot 15, Cellranger 5.0.1, default parameters) to generate consensus annotation files for each sample. Using the scRepertoire[51] package, sample-specific consensus annotation files were consolidated into a list of TCR sequencing results and then integrated with the Seurat object, containing RA8–15, using the combineExpression function. Clonotypes were called based on the CDR3 nucleotide sequence and VDJC gene (i.e., scRepertoire nomenclature "gene+nt") and clonotype expansion were assigned based on the relative frequency. The interconnectivity between specific cell types, both within and between tissue compartments, were visualized using chord diagrams.

**Reporting summary**. Further information on research design and data analysis is available in the Nature Research Reporting Summary linked to this article.

## Data availability
Sequence data have been deposited at the European Genome-phenome Archive (EGA), which is hosted by the EBI and the CRG, under accession number EGAS00001005241 (Dataset accession number: EGAD00001007586 https://ega-archive.org/studies/EGAS00001005241). Further information about EGA can be found on https://ega-archive.org "The European Genome-phenome Archive of human data consented for biomedical research". All other data are available in the article and its Supplementary files or from the corresponding author upon reasonable request. Source data are provided with this paper.

## Code availability
Codes are available at https://github.com/PfizerRD/Argyriou_Wadsworth_etal_2022.

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

## Acknowledgements

We thank the patients who donated samples and the medical staff at the Rheumatology Clinic of Karolinska University Hospital. Julia Boström, Gloria Rostvall, and Susana Hernandez Machado are acknowledged for organizing the sampling, storage, and administration of biomaterial. We thank Leonid Padyukov and Barbro Larsson for HLA-DR genotyping of the samples and Lena Israelsson for technical assistance with ACPA measurement. We thank Juan Sebastian Diaz Boada for bioinformatic technical assistance. Figure 6f was created using BioRender.com. This study is supported by grants from Dr. Margaretha Nilssons (K.C.), the Nanna Svartz (K.C., 2020-00349), the Börje Dahlin (K.C.) and the Ulla and Gustaf af Ugglas foundations (K.C., 2018-02627) as well as the Swedish association against rheumatism (K.C., R-969294). This project has received funding from the Innovative Medicines Initiative 2 Joint Undertaking (JU) under grant agreement No 777537 (RTCure) (V.M.). The JU receives support from the European Union's Horizon 2020 research and innovation programme and EFPIA.

## Author contributions

A.A., A.L., and C.G. performed and analyzed the flow cytometry experiments. M.H.W. performed the 10X experiments and led the single-cell analysis. S.M.C. performed preliminary 10X experiments and single-cell data analysis. A.H.H. diagnosed RA patients and collected clinical information. A.v.V. performed flow cytometry sorting. K.K. contributed to and oversaw the design, analysis, and interpretation of the single-cell experiments. A.W. contributed to the design of the study. V.M. contributed to the design of the study and the RA sample collection. A.A., M.H.W., and K.C. generated the figures. K.C designed the study, oversaw the analyses, and wrote the paper. All authors read, edited, and approved the paper.

## Funding

## Competing interests

A.A., A.L., C.G., A.H.H., A.vV., V.M., and K.C. declare no competing interests. M.H.W., S.M.C., K.K., and A.W. are employees of Pfizer, Inc, Cambridge, MA 02139, United States.
