## [Peer Review File · Nature Communications]

Single cell sequencing identifies clonally expanded synovial CD4+ TPH cells expressing GPR56 in rheumatoid arthritisREVIEWER COMMENTS

Reviewer #1 (Remarks to the Author):

The paper by Argyriou et al focuses on CD4+ T cells in the synovial fluid of patients with ACPA+ rheumatoid arthritis (RA). Particularly, they characterize in depth the Tph subset and the CD4+ cytotoxic T cells in the synovial fluid of these patients. By analysing TCR sequences in the blood and SF of ACPA+ RA patients, they find that the most expanded clones in the SF belong to CD4 cytotoxic or Tph subsets, and find shared TCR sequences in these two subpopulations, which they interpret as indicative of common differentiation.

While it is not the first manuscript with detailed single cell transcriptomics of the synovial compartment, this paper separates from the previous ones because of its focus on the CD4+ T cell compartment. The existence of Tph in the synovial fluid of patients with RA had already been reported, however, such a comprehensive immune profiling of the CD4+ T cell compartment combining gene expression and clonality had not been performed. Furthermore, the authors identify GPR56 as a marker for Tph. This is of importance, since until now there was no exclusive surface marker for this cell type.

The experiments shown in this manuscript map the CD4 T cell subsets in SF in RA in great detail, however, it is purely descriptive. The assumption that there might be a common differentiation or the conversion of Tph into CD4 cytotoxic cells (as depicted in Figure 6) is based on shared CDR3 sequences, but no functional data is provided.

Specific comments:

1) Is GPR56 a marker also for Tph cells in ACPA- RA patients? This could be easily addressed by flow cytometry. Even if less abundant, are GPR56+ Tph cells also found in the SF of ACPA- patients? Are PRDM1+ Tph and CXCL13+ Tph found in ACPA- patients?

2) GPR56 in SF seems exclusive of PD-1 hi cells (Fig 3), while cytotoxic CD4 cells in the SF do not express GPR56 at protein level, even though ADGRG1 is expressed also in the SF. The authors conclude that SF cytotoxic CD4 cells do not express GPR56. Is the level of gene expression different in the Tph and CD4 cytotoxic cells, at least so much that it could explain the differences in protein expression at the membrane? Could GPR56 be shed from the cell surface as it happens with NK cells? Do peripheral CD4 cytotoxic cells lose GPR56 after activation?

3) The authors claim that shared CDR3 sequences among Tph cells (both states) and CD4 cytotoxic cells could represent a common origin. The idea is attractive, but the data as it is presented does not support it: in figure 5 it is very difficult to follow which clones are shared in the different clusters. The figure could be maintained for a global view, but a table with detailed information on how many cells in each of the clusters share which CDR3 sequences is necessary to support this claim. At this point the information in the text is vague:

"Some of the expanded clones were shared between PB and SF"

"In SF, the CXCL13+ TPH cluster shared TCR sequences with the PRDM1+ TPH, Humanin+ CD4+ and proliferating CD4+ T-cell clusters"

"Common CDR3s were also identified between cytotoxic CD4+ T cells and CXCL13+ TPH cells"

"We observed that some CDR3 sequences were shared between cytotoxic CD4+ T cells in PB and SF".

"Similarly, CXCL13+ TPH cells had common CDR3s in PB and SF"

I could not follow if the shared sequences in the different clusters is a general phenomenon of all (or most) expanded clones, or rather a very isolated feature of a few clones.

4) The authors suggest that PRDM1+ Tph CD4+ T cells could differentiate into cytokine producing CXCL13+ Tph cells, and later into cytotoxic cells (Figure 6), based on the existence of an undetermined number of shared CDR3 sequences. Can the authors provide in vitro evidence for

these steps of conversion? i.e. specific changes in cytokine profile, up- or downregulation of transcription factors

Minor points:

The authors perform scRNA analysis in PB and SF of ACPA+ RA patients. Tph cells had been found more abundant in synovial fluid of patients with high leukocyte counts in SF (Zhang et al. 2019). Are these two classifications identifying a similar group of patients? A comment in the discussion would be helpful.

Some of the RA patients are under corticosteroid treatment. Is the treatment affecting the RNA signature/clonal expansion in these donors?

Reviewer #2 (Remarks to the Author):

In this manuscript, Argyriou et al used single cell sequencing and multi-parameter flow cytometry to identify distinct subsets of CD4+ T cells in PBMC and synovial fluid of ACPA positive and negative RA patients. Their study verified the GPR56 as a marker of circulating cytotoxic cells, delineated the synovial TPH CD4+ T-cell subsets, and illustrated the Trm marker expressed on TPH cells.

This dataset is of high interest to the field. However, there are some important limitations to the analysis and interpretation that limit the interest in this study.

Major comments

1. In this study, the author isolated "CD4+ T cells" through single CD4 positive selection directly from whole PBMC or synovial fluid. Because some monocytes and macrophages also express CD4, the author should show the purity test by flow cytometry before single cell sequencing, or remove the possible contamination during the data analysis.

2. This study recruited patients treated with or without steroid or immune suppressants. It is important to compare the CD4 T cells clusters between these groups to clarify the differences are disease specific or drug induced.

3. The authors conclude that "The presence of CD4+ CTL only in SF of ACPA+ RA suggest that they have a pathogenic function". It is way too subjective. To make this conclusion, functional analysis should have been carried out to understand the contribution of these cells in RA.

4. RNA velocity or cytotracer analysis should be applied to the differentiation hypothesis between TPH and CD4 CTL cells. Further in vitro differentiation assay is recommended.

5. We would prefer to rename the cluster according to their cluster size, which gave us the rank information of cluster size. The author can either rename the cluster according to their function. Besides that, you only described cluster 15, 10, 13, 14, 3 and 12, what happened to other clusters?

6. About the method part in single cell processing, did the author mean that CITE-Seq hashes failed in patient 3? Could the author provide a more specific list showing the treatment of each patient in single-cell sequence dataset? The treatment includes sequence approaches and the analysis software, such as harmony or CCA.

7. Line 144 & 298, to verify GPR56 as a marker of TPH, authors should compare the GPR56 expression in TPH cell and non-TPH cell.

8. Line 162, Supplementary Figure 7 is not related to the description.

9. Line 246, the data showed no significant difference between the cytotoxic molecule expressions in CD8 T cells (Supplementary Fig. 2.), so the conclusion, cytotoxic CD8+ T cells also tended to be more cytotoxic in ACPA+ RA and could contribute to a perforin-dependent citrullination mechanism, is not reliable.

10. Line 256, Hobit is one of the gene highly expressed in CD4+ CTL, and Hobit maintain the Gzmb expression in CD8 T cells as the reference reported. This evidence is not sufficient to conclude that

'Hobit also drive CD4+ CTL differentiation in ACPA+ RA'.

11. Current study and analysis do not give enough support for Figure 6.

12. In supplementary Figure 2a, the representative flow cytometry dot staining of GZMB and PRF1 are same.

Minor comments

1. line 85, what is the full name of SFMC?

2. Start from line 108, the process of annotating the cluster type should be more detailed.

3. In line 128, it should be "In PB, ADGRG1 was expressed in TPH cells".

4. Line 317, should be 1×10^6 .

5. The naming of cluster 2, CD4+ T cells, is not rational. Because all cells are CD4+ T cells.

6. Line 452-453, The Fig 3c legend does not match the figure.

7. Supplementary Figure 8b, the y axes are all wrong.

8. The genes mentioned from line 112 to line 121 deserve feature plots in the supplementary figure.

Reviewer #3 (Remarks to the Author):

In this manuscript by Argyriou et al., different populations of cytotoxic and peripheral helper T cell subsets are characterized by flow cytometry, single-cell RNA-sequencing and single-cell TCR sequencing in blood and synovial fluid of RA patients. The authors show that specific subpopulations of Tph and cytotoxic CD4 T cells are enriched and expanded in the SF of ACPA-positive RA patients, in specific the GPR56+ Tph population that also shows characteristics of tissue residency. The role of cytotoxic CD4 in autoimmunity is a timely topic and the extensive set of data provide important new insights in the phenotype and possible function of these cells in local inflammation. Please find comments below.

General comments:

One important aspect that is now missing is the relation between the identified cell subsets. In the discussion the authors postulate differentiation pathways, but they do not provide the pseudo-time data. These analysis would certainly strengthen the data and conclusions. For some of the key (functional) characteristics of the T cell subsets, protein confirmation is missing and should be added (e.g. CXCL13 and Blimp-1). Regarding the TCR sequencing data, pooling of the data may have led to a false interpretation of overlapping clone data.

Specific comments

Fig 2. If I understand correctly, CD4 T cells from peripheral blood were sorted on total CD4 T cells, including naïve T cells. Since the CD4 cytotoxic and Tph populations have a memory phenotype, the proportion calculation in Fig 2D may be somewhat distorted due to the relatively high percentage of naïve CD4 T cells in PB, especially since the percentages in SF show also quite some variation. It would be insightful to also calculate the percentages of these subsets within the CD4 memory T cell fraction by excluding the naïve T cell cluster from the analysis. The heterogeneity of the data may be related to patient characteristics, in specific to the treatment they received. This should be discussed.

Fig 2. In line 118 cluster 4 is identified as cytotoxic based on the expression of NKG7, GZMH and PRF1. In line 130 HOBIT and GZMB are also mentioned. It would be good to keep this consistent. HOBIT is also missing in the heatmap Fig. 2c, and GZMB seems to be mostly enriched in cluster 8.

Fig 2c. 'Heatmap showing gene expression values for a curated list of CD4 T cell subset-defining genes'; information on p-value, log-fold change, cut-off value and the source(s) the selection was based on are missing.

It would be good to provide a table with the top genes defining the different clusters (for SF and

PB) in the main figures.

Line 119. Cluster 4 is also present in PB, this should be addressed.

Lines 120-122. Please give the frequency range also for these clusters.

Fig 3. To show that the GPR56+ cells in SF differ from those in the blood it would be good to also show the (lack of) expression of PD-1, CXCL13, MAF, ICOS etc in parallel for the PB T cells.

Fig 3b. Is there a significant difference between ACPA+ and ACPA- regarding the expression of PD-1 and MHCII if you compare all individuals?

Fig 4. Since CXCL13 and PDRM1 are defining the Tph clusters, this should also be confirmed on the protein level (co-expression with GPR56) as is shown for other markers.

Line 159. The authors conclude that the cytotoxic CD4 T cells are recirculating and the Tph not. Nevertheless Tph cells are also found in circulation and also show some overlap in TCR. So although these cells do not express tissue resident markers, there may be (re)circulating.

Fig 4c. When addressing tissue residency, CD69 expression is indeed one of the hallmarks. Therefore it would be informative to not only show the MFI but also the percentage expressing cells within the populations.

TCR data. In what percentage of T cells a TCR beta and/or alpha chain could be detected? In figure 5a, 7 out of 10 clones represent the beta chain. Is this because of a difference in expansion or a difference in sequence recovery? Also, with 58bp the reads are rather short. Very low efficiency may introduce bias in the data, please comment.

Fig 5b. The most expanded clones seem to be present within the cytotoxic CD4 T cell population, although two patients (RA#2 and RA#7) show a completely different pattern. It would be good to mention this in the text.

Fig 5c-d. If I understand correctly, in these figures all sequences from different patients are pooled. Since the full HLA background of the patients is unknown, it does not seem to be fair to draw conclusions about shared clones between subsets and between PB-SF. This would be more relevant on the individual level (so is there TCR overlap between subsets or between PB and SF within a patient).

For the level of overlap it would be good to mention percentages.

Lines 181-183. and supplementary Fig 9. The authors state that identical clones are shared between individuals. However, only the beta chain sequence is shown to be overlapping and therefore the TCR may be different between these individuals. Also, why was a cut-off of 4 used in this setting?

Discussion. The authors postulate that the Tph clusters correspond to two states of differentiation (lines 222-225) and that Tph cells lead to cytotoxic CD4 T cell expanded clones (lines 274-276). By performing a pseudotime analysis, the authors can demonstrate whether this is indeed the most likely connection. This would further strengthen the data.

Minor comments

Please use T-cell or T cell consistently

Line 160. this statement refers to SF, please add.

General comments:

We thank the reviewers for their constructive suggestions and comments. We have addressed all comments and feel that the manuscript has improved. To answer some of the comments, we have generated new single cell analysis experiments on peripheral blood and synovial fluid samples from ACPA- and matched ACPA+ RA samples (n=4 in each group). We have addressed all comments by 1) validating the T_{PH} phenotypes by flow cytometry 2) performing in vitro activation of sorted T_{PH} cells 3) generating RNA velocity analysis 4) refining the CDR3 sequences analyses. All these experiments support a common differentiation pathway between the two T_{PH} subsets and effector CD4⁺ T cells. We also found that cytotoxic CD4⁺ T cells are differentiating from a different branch directly from effector CD4⁺ T cells. In this dataset, we identify most of the SF expanded clones in CXCL13^{high} T_{PH} cells and propose therefore the new title: “**Single cell sequencing reveals clonally expanded CXCL13^{high} peripheral helper CD4⁺ T cells expressing the G protein-coupled receptor GPR56 in synovial fluid of RA patients**”. All manuscript modifications are highlighted by track changes in the manuscript “with_Track_changes_Argyriou_Wadsworth_20210519” and a cleaned version is available in: “resubmission_Argyriou_Wadsworth_20220214”.

Response to reviewer comments:

Reviewer 1

The paper by Argyriou et al focuses on CD4⁺ T cells in the synovial fluid of patients with ACPA+ rheumatoid arthritis (RA). Particularly, they characterize in depth the Tph subset and the CD4⁺ cytotoxic T cells in the synovial fluid of these patients. By analysing TCR sequences in the blood and SF of ACPA+ RA patients, they find that the most expanded clones in the SF belong to CD4 cytotoxic or Tph subsets, and find shared TCR sequences in these two subpopulations, which they interpret as indicative of common differentiation.

While it is not the first manuscript with detailed single cell transcriptomics of the synovial compartment, this paper separates from the previous ones because of its focus on the CD4⁺ T cell compartment. The existence of Tph in the synovial fluid of patients with RA had already been reported, however, such a comprehensive immune profiling of the CD4⁺ T cell compartment combining gene expression and clonality had not been performed.

Furthermore, the authors identify GPR56 as a marker for Tph. This is of importance, since until now there was no exclusive surface marker for this cell type.

The experiments shown in this manuscript map the CD4 T cell subsets in SF in RA in great detail, however, it is purely descriptive. The assumption that there might be a common differentiation or the conversion of Tph into CD4 cytotoxic cells (as depicted in Figure 6) is based on shared CDR3 sequences, but no functional data is provided

We thank reviewer 1 for her/his thorough analysis and positive feedback. Based on reviewer 1’s suggestions, we have now performed additional single sequencing experiments as well as functional experiments as outlined below to strengthen our hypothesis.

1-a) Is GPR56 a marker also for T_{PH} cells in ACPA- RA patients? This could be easily addressed by flow cytometry. Even if less abundant, are GPR56+ T_{PH} cells also found in the SF of ACPA- patients?

We have analysed GPR56 expression on T_{PH} (PD-1^{high}) versus non T_{PH} cells (PD-1⁻) on SF CD4⁺ T cells in ACPA- and ACPA+ patients by flow cytometry (See also comment 7 from reviewer 2). This analysis (Fig. 3b and below) shows that GPR56 is also expressed on T_{PH} cells in SF of ACPA- RA patients although to a lesser extent than in ACPA+ RA patients (p=0.0007).

Representative flow cytometry dot plots of GPR56 expression within PD-1^{high} (T_{PH}) and PD-1⁻ (non-T_{PH}) CD4⁺ T cells in ACPA+ (upper panel) and ACPA- (lower panel) RA SF, quantified in n=9 ACPA- SF and n=12 ACPA+ SF. Line represents median, two-tailed Mann-Whitney U test. Blue dots indicate ACPA- RA SF and red dots indicate ACPA+ RA SF.

1-b) Are PRDM1+ T_{PH} and CXCL13+ T_{PH} found in ACPA- patients?

In the revised manuscript, we also present single cell data from 4 ACPA- RA patients. We indeed identify the presence of the two T_{PH} subsets in ACPA- RA SF (see below and Fig. 2 and supp Fig. 6). Based on discrepancies between PRDM1 and BLIMP-1 expression, we have renamed these subsets: CXCL13^{low} T_{PH} (previously PRDM1+ T_{PH}) and CXCL13^{high} T_{PH} (previously CXCL13+ T_{PH}). All other proteins tested (CXCL13, PD-1, MHCII, GPR56, LAG3) correlated with the single cell RNA sequencing data. (See result section page 8, line 157-171 and discussion section page 14, line 309-319). Of note, we also found that LAG-3 expression was lower on ACPA- T_{PH} as compared to ACPA+ (Fig. 4c).

Frequencies of CD4⁺ T cells clusters in SF from n=8 RA patients (4 ACPA+, 4 ACPA-). Red rectangles highlight T_{PH} subsets. Circle indicates ACPA+ patients, triangle indicates ACPA- patients. Line represents median, two-tailed Mann-Whitney U test.

To evaluate the frequency of these cells by flow cytometry, we took advantage of the differential GPR56 expression on these two T_{PH} subsets (**Fig. 3a, c** and **Supp. Fig. 11a**) and evaluated their frequencies based on PD-1 and GPR56 expression (**See below and Fig. 3c**). We also investigated the expression of PD-1, MHCII, CXCL13 and Blimp-1 in these two subsets in ACPA- and ACPA+ SF (see comment 9 from reviewer 3). We observe that, although ACPA- T_{PH} cells are less frequent than in ACPA+ SF, both subsets show similar characteristics of PD-1, MHC-II, CXCL13 and Blimp-1 expression as their ACPA+ counterparts. This data is an important information as it suggests that the transition from CXCL13^{low} T_{PH} towards CXCL13^{high} T_{PH} is enhanced in ACPA+ RA individuals.

Representative flow cytometry dot plots of the two T_{PH} states (PD-1^{high} GPR56^{low} and PD-1^{high} GPR56⁺) and quantification of their frequency, MHCII and PD-1 expression in n=9 ACPA- SF and n=11-12 ACPA+ SF. CXCL13 and BLIMP-1 expression within PD-1-GPR56- (non-T_{PH}) and PD-1^{high} GPR56^{low} and PD-1^{high} GPR56⁺ (2 T_{PH} states) in CD4⁺ T cells in ACPA- and ACPA+ RA SF in n=5 ACPA- SF and n=5 ACPA+ SF. Line represents median, two-tailed Mann-Whitney U test. Red dots indicate ACPA+, blue dots indicate ACPA- RA patients.

2-a) *GPR56 in SF seems exclusive of PD-1 hi cells (Fig 3), while cytotoxic CD4 cells in the SF do not express GPR56 at protein level, even though ADGRG1 is expressed also in the SF. The authors conclude that SF cytotoxic CD4 cells do not express GPR56. Is the level of gene expression different in the TPH and CD4 cytotoxic cells, at least so much that it could explain the differences in protein expression at the membrane?*

We have analysed *ADGRG1* expression in cytotoxic CD4⁺ T cells and CXCL13^{high}T_{PH} cells in ACPA+ SF and we observe indeed significantly lower *ADGRG1* expression in cytotoxic CD4⁺ T cells as compared to CXCL13^{high} T_{PH} cells (**See below and Supp Fig. 10c**).

***ADGRG1* expression in CXCL13^{high} T_{PH} versus cytotoxic CD4⁺ T cells in ACPA+ SF (n=4 RA patients)**

2-b) *Could GPR56 be shed from the cell surface as it happens with NK cells? Do peripheral CD4 cytotoxic cells lose GPR56 after activation?*

We have tested this hypothesis by activating SFMC and PBMC with either CD3/CD28-beads or PMA/ionomycin for 3 hours. While CD3/CD28 beads had no effect on GPR56 expression, PMA/ionomycin induced GPR56 down-regulation in a similar manner as reported for NK cells (**below and Supp Fig. 13**). This decrease was observed for both GPR56⁺ T_{PH} (SF) and peripheral cytotoxic GPR56⁺ CD4⁺ T cells. It is therefore plausible that, in an inflammatory milieu, GPR56 is endocytosed or cleaved from CD4⁺ T cell surface contributing to GPR56

downregulation on cytotoxic CD4⁺ T cells in SF. This data is presented in result section page 9 (line 174-182) and discussed page 17 (line 370-374). However, it is intriguing that T_{PH} cells retain high GPR56 level in SF and it will require further investigation.

GPR56 downregulation after activation. (a) GPR56 expression on peripheral blood CD4⁺ T cells after 3 hours activation with CD3/CD28 beads or PMA/Ionomycin (n=3 ACPA-, n=3 ACPA+ RA patients). (b) GPR56 expression on synovial CD4⁺ T cells after 3 hours activation with CD3/CD28 beads or PMA/Ionomycin in 6 RA patients (n=4 ACPA-, n=4 ACPA+ RA patients). Line indicates median, two-tailed Wilcoxon paired test. ns: not significant. Blue dots indicate ACPA- and red dots indicate ACPA+ RA patients.

3) The authors claim that shared CDR3 sequences among TPH cells (both states) and CD4 cytotoxic cells could represent a common origin. The idea is attractive, but the data as it is presented does not support it: in figure 5 it is very difficult to follow which clones are shared in the different clusters. The figure could be maintained for a global view, but a table with detailed information on how many cells in each of the clusters share which CDR3 sequences is necessary to support this claim. At this point the information in the text is vague:

- “Some of the expanded clones were shared between PB and SF”
- “In SF, the CXCL13⁺ TPH cluster shared TCR sequences with the PRDM1⁺ TPH, Humanin⁺ CD4⁺ and proliferating CD4⁺ T-cell clusters”
- “Common CDR3s were also identified between cytotoxic CD4⁺ T cells and CXCL13⁺ TPH cells”
- “We observed that some CDR3 sequences were shared between cytotoxic CD4⁺ T cells in PB and SF”.
- “Similarly, CXCL13⁺ TPH cells had common CDR3s in PB and SF”

I could not follow if the shared sequences in the different clusters is a general phenomenon of all (or most) expanded clones, or rather a very isolated feature of a few clones.

We agree with this comment (see also comments 12-15 from reviewer 3). In the revised **Figure 6b** and **below b**), we are now presenting shared CDR3 sequences between all cells (including clones) in the different clusters. The CDR3 overlap is quantified using a jaccard index. Moreover, we also analysed the sharing of CDR3 clones by focusing on expanded clones (n≥2) in each cluster per patient (**Figure 6c** and **below c**). Finally, we provide raw information on shared CDR3 in expanded clones in **supplementary table 7**). In SF, CXCL13^{high} T_{PH} cells show clonal overlap with CXCL13^{low} T_{PH} (Jaccard Index (JI):0.038), proliferating (JI:0.031), effector (JI:0.023) and cytotoxic CD4⁺ T cells (JI:0.013). We then focused our analysis on expanded clones in CXCL13^{high} T_{PH} CD4⁺ T cells (n≥2). A median of 59% of the CXCL13^{high} T_{PH} expanded clones shared CDR3 sequences with cells from other clusters (either clones or unique cells) (**Supp. Table 6d**). These shared sequences were identified in all patients. 28% of

these clones were shared with proliferating CD4⁺ T cells, 25% with CXCL13^{low} T_{PH}, 19% with effector CD4⁺ T cells, and 6% with cytotoxic CD4⁺ T cells (**Fig. 6c, Suppl. Table 6e**).

Altogether, we confirmed a sharing of CDR3 sequences between CXCL13^{high} T_{PH} expanded clones and CXCL13^{low} T_{PH} cells (clones or unique cells) as well as effector CD4⁺ T cells and to a lesser extend cytotoxic CD4⁺ T cells. This data is presented page 11-12 and discussed page 14 (line 301-303).

b) Jaccard overlap quantifications for clonotypes between cell clusters across compartments. **c)** % of TCR clonal overlap between expanded T cell clones (n ≥ 2 cells) and T cells from other clusters (n ≥ 1 cell).

4) *The authors suggest that PRDM1+ TPH CD4+ T cells could differentiate into cytokine producing CXCL13+ TPH cells, and later into cytotoxic cells (Figure 6), based on the existence of an undetermined number of shared CDR3 sequences. Can the authors provide in vitro evidence for these steps of conversion? i.e. specific changes in cytokine profile, up- or downregulation of transcription factors*

To evaluate the conversion from CXCL13^{low} T_{PH} into CXCL13^{high} T_{PH}, we sorted GPR56^{low} PD-1^{high} CD4⁺ T cells and GPR56^{high} PD-1^{high} CD4⁺ T cells from ACPA+ RA SF and submitted these cells to 48 hours CD3/CD28 bead activation. We then evaluated the expression of PD-1, GPR56, CXCL13, BLIMP-1 and Perforin-1 (**Figure 6d and below**). We observed that GPR56^{low} T_{PH} cells upregulated PD-1 to a level comparable to CXCL13^{high} T_{PH} (p=0.0005). We also observed an upregulation of GPR56 and BLIMP-1 in all 3 patients tested which did not reach significance due to lack of power. Although our initial plan was to perform n=6 experiments, we could not recover enough PD-1^{high} GPR56^{low} T_{PH} cells in 3 of the tested samples. Finally, we did not observe any striking upregulation of perforin-1 expression. Altogether, these data shows that CXCL13^{low} T_{PH} can convert into CXCL13^{high} T_{PH} but not in cytotoxic CD4⁺ T cells in the tested conditions. This data is presented page 12 (line 258-265) and discussed page 14 (line 303-308). Of note, we didn't see any increase in CXCL-13 production in the 2 T_{PH} subsets after T cell activation which might be due to the lack of additional stimulation factors as discussed page 12 line 263-265.

Activation of T_H subsets. Representative flow cytometry dot plots showing the expression of PD-1, GPR56, CXCL13, BLIMP-1, and PERF-1 in control (upper panel) or 48hours CD3/CD28 activated T_H cell states (lower panel) (PD-1^{high} GPR56^{low} and PD-1^{high} GPR56⁺ from ACPA+ SF), quantified in n=3 ACPA+ RA patients. (d) Data are from a pool of three independent experiments where a circle is a single replicate. Line indicates median, two-tailed Wilcoxon paired test.

Further, to explore the connection between CD4⁺ T cell subsets (see also comment 4 from reviewer 2 and comment 16 from reviewer 3), we performed velocity analysis on the sc dataset (**Figure 6e** and **below**). We indeed observe a transition from CXCL13^{low} T_H into CXCL13^{high} T_H which was more obvious in ACPA+ CD4⁺ T cells (lower panel). Interestingly, effector CD4⁺ T cells and SESN3⁺ CD4⁺ T cells project into two independent branches leading to T_H in one direction and cytotoxic CD4⁺ T cells in the other direction. These observations suggest that, in RA, T_H and cytotoxic CD4⁺ T cells are differentiating from effector T cells through two independent pathways. The sharing of CDR3 sequences that we observe between these two subsets could be due to common progenitors present in the effector CD4⁺ T cell subset populations. This data is presented page 12 (line 265-276) and discussed in page 14 (Line 304-308), page 15 (Line 344), page 16 (345-351).

Velocity plot showing the connection between the different CD4⁺ T cell clusters in RA split in ACPA- (upper panel, n=4) and ACPA+ (lower panel, n=11).

Minor points:

5) *The authors perform scRNA analysis in PB and SF of ACPA+ RA patients. TPH cells had been found more abundant in synovial fluid of patients with high leukocyte counts in SF (Zhang et al. 2019). Are these two classifications identifying a similar group of patients? A comment in the discussion would be helpful.*

This is an interesting point. Zhang et al. showed that T_{PH} cells are more abundant in leukocyte-rich synovial tissues from RA patients. Our study was performed on SFMC isolated from synovial fluid and hence we cannot evaluate the aspect of tissue-leucocyte enrichment here. To perform the flow cytometry and scRNA experiments on sorted CD4⁺ T cells, samples with a reasonable SFMC number ($\geq 10 \times 10^6$ cells) were selected. We have discussed this aspect in discussion page 14 (line 319-), page 15 (line 320-329).

6) *Some of the RA patients are under corticosteroid treatment. Is the treatment affecting the RNA signature/clonal expansion in these donors?*

We have evaluated the frequencies of CD4⁺ T cell clusters identified in SF by single cell sequencing and flow cytometry depending on the treatment by prednisolone, methotrexate or biologics at time of sampling (see also comment 2 from reviewer 2 and comment 2 from reviewer 3). This analysis is presented in **Supp. Fig. 7**. We did not detect any significant difference but cannot rule out the possibility that treatment will affect CD4⁺ T cell subsets in the joint. We have therefore commented this possibility in discussion page 14/15 (319-329).

Reviewer 2

In this manuscript, Argyriou et al used single cell sequencing and multi-parameter flow cytometry to identify distinct subsets of CD4⁺ T cells in PBMC and synovial fluid of ACPA positive and negative RA patients. Their study verified the GPR56 as a marker of circulating cytotoxic cells, delineated the synovial TPH CD4⁺ T-cell subsets, and illustrated the Trm marker expressed on TPH cells.

This dataset is of high interest to the field. However, there are some important limitations to the analysis and interpretation that limit the interest in this study.

We thank reviewer 2 for highlighting the importance of this study. We have performed additional functional experiments and analysis to validate our findings.

Major comments

1. In this study, the author isolated “CD4⁺ T cells” through single CD4 positive selection directly from whole PBMC or synovial fluid. Because some monocytes and macrophages also express CD4, the author should show the purity test by flow cytometry before single cell sequencing, or remove the possible contamination during the data analysis.

We have performed new single cell sequencing experiments (PB and SF from n=4 ACPA- and 4 ACPA+ RA patients) and have filtered the dataset for CD14, CD8a, CD8b and MS4A1. Finally, to further remove non-T cells that passed our initial screen, clusters with (1) low expression of CD3, (2) differentially expressed genes (DEGs) that are known markers of non-T Cells and (3) high enrichment scores for non-T Cell gene modules (MSigDB) were removed from the dataset (method section page, 20 (line 466-468), page 22 line 496-499).

2. This study recruited patients treated with or without steroid or immune suppressants. It is important to compare the CD4 T cells clusters between these groups to clarify the differences are disease specific or drug induced.

We have evaluated the frequencies of CD4⁺ T cell clusters identified in SF by single cell sequencing and flow cytometry depending on the treatment by prednisolone, methotrexate, or biologics at time of sampling (see also comment 6 from reviewer 1 and comment 2 from reviewer 3). This analysis is presented in **Supp. Fig. 7**. We did not detect any significant difference but cannot rule out the possibility that treatment will affect CD4⁺ T cell subsets in the joint. We have therefore commented this possibility in discussion page 14/15 (319-329).

3. The authors conclude that “The presence of CD4⁺ CTL only in SF of ACPA+ RA suggest that they have a pathogenic function”. It is way too subjective. To make this conclusion, functional analysis should have been carried out to understand the contribution of these cells in RA.

We agree with reviewer 2 that this sentence is not supported by functional experiments and was therefore removed from Discussion section.

4. RNA velocity or cytotracer analysis should be applied to the differentiation hypothesis between TPH and CD4 CTL cells. Further in vitro differentiation assay is recommended.

We totally agree with this suggestion (see also comment 4 from reviewer 1 and comment 16 from reviewer 3). To evaluate the conversion from CXCL13^{low} T_{PH} into CXCL13^{high} T_{PH}, we sorted GPR56^{low} PD-1^{high} CD4⁺ T cells and GPR56^{high} PD-1^{high} CD4⁺ T cells from ACPA+ RA SF and submitted these cells to 48 hours CD3/CD28 bead activation. We then evaluated the

expression of PD-1, GPR56, CXCL13, BLIMP-1 and Perforin-1 (**Figure 6d and below**). We observed that GPR56^{low} T_{PH} cells upregulated PD-1 to a level comparable to CXCL13^{high} T_{PH} (p=0.0005). We also observed an upregulation of GPR56 and BLIMP-1 in all 3 patients tested which did not reach significance due to lack of power. Although our initial plan was to perform n=6 experiments, we could not recover enough PD-1^{high} GPR56^{low} T_{PH} cells in 3 of the tested samples. Finally, we did not observe any striking upregulation of perforin-1 expression. Altogether, these data shows that CXCL13^{low} T_{PH} can convert into CXCL13^{high} T_{PH} but not in cytotoxic CD4⁺ T cells in the tested conditions. This data is presented page 12 (line 258-265) and discussed page 14 (line 303-308). Of note, we didn't see any increase in CXCL-13 production in the 2 T_{PH} subsets after T cell activation which might be due to the lack of additional stimulation factors as discussed page 12 line 263-265.

Activation of T_{PH} subsets. Representative flow cytometry dot plots showing the expression of PD-1, GPR56, CXCL13, BLIMP-1, and PERF-1 in control (upper panel) or 48hours CD3/CD28 activated T_{PH} cell states (lower panel) (PD-1^{high} GPR56^{low} and PD-1^{high} GPR56⁺ from ACPA+SF), quantified in n=3 ACPA+ RA patients. Data are from a pool of three independent experiments where a circle is a single replicate. Line indicates median, two-tailed Wilcoxon paired test.

Further, to explore the connection between CD4⁺ T cell subsets (see also comment 4 from reviewer 1 and comment 16 from reviewer 3), we performed velocity analysis on the sc dataset (**Figure 6e and below**). We indeed observe a transition from CXCL13^{low} T_{PH} into CXCL13^{high} T_{PH} which was more obvious in ACPA+ CD4⁺ T cells (lower panel). Interestingly, effector CD4⁺ T cells and SESN3⁺ CD4⁺ T cells project into two independent branches leading to T_{PH} in one direction and cytotoxic CD4⁺ T cells in the other direction. These observations suggest that, in RA, T_{PH} and cytotoxic CD4⁺ T cells are differentiating from effector T cells through two independent pathways. The sharing of CDR3 sequences that we observe between these two subsets could be due to common progenitors present in the effector CD4⁺ T cell subset populations. This data is presented page 12 (line 265-276) and discussed in page 14 (Line 304-308), page 15 (Line 344), page 16 (345-351).

Velocity plot showing the connection between the different CD4+ T cell clusters in RA split in ACPA- (upper panel, n=4) and ACPA+ (lower panel, n=11).

5. We would prefer to rename the cluster according to their cluster size, which gave us the rank information of cluster size. The author can either rename the cluster according to their function. Besides that, you only described cluster 15, 10, 13, 14, 3 and 12, what happened to other clusters?

We have renamed the clusters based on cluster size. We also provide a detailed description for each cluster in page 6 (line 97-112), **Supp. Fig. 5c-d**.

6. About the method part in single cell processing, did the author mean that CITE-Seq hashes failed in patient 3? Could the author provide a more specific list showing the treatment of each patient in single-cell sequence dataset? The treatment includes sequence approaches and the analysis software, such as harmony or CCA.

We apologize that our methods section was unclear. To elaborate, we could not deconvolute the multiplexed PBMC and SFMC based on CITE-Seq hashes in the first single cell experiments, as the result of sub-optimal tagging (RA1-7; 10X-1 experiments). Instead, to interpret these precious patient samples in our original submission, we generated a compartment-specific reference using PBMC and SFMC samples from two more volunteers, and then reference-mapped our mixed samples to assign PB and SF compartment membership (Hao Y. et al. Cell, 2021). While this method is widely used to successfully assign unknown cell types (CITE), our application for tissue deconvolution is novel and, as such, could introduce uncertainty.

We thank the reviewer for this useful comment and to completely remove this uncertainty from our analysis, we acquired samples from 8 more patients and generated both PBMC and SFMC unambiguously in additional single cell experiments (RA8-15; 10X-2 experiments). In the revised manuscript, because we do not correct for compartment-specific effects (which is the only uncertainty in RA1-7), we used all patient samples for the unsupervised analysis, cell type annotation, and RNA Velocity analysis (**See below and Supp Figure 3, Supp Table X**). However, for the TCR analysis, we only used data generated from RA8-15 to address our questions related to clonality within and across tissue compartments. All legends include the number of samples included in the analysis and supplementary Table 1 describes which samples have been assigned to which experiment. See also method section page 19 (line 424-445).

UMAP displaying 12 CD4⁺ T cell clusters in combined SF and PB (left panel, n=15 RA (11 ACPA+, 4 ACPA-), n=166383 cells), SF (middle panel, n=8 RA (4 ACPA+, 4 ACPA-), n=305288 cells) and PB (right panel, n=8 RA (4 ACPA+, 4 ACPA-), n=481677 cells).

7.Line 144 & 298, to verify GPR56 as a marker of T_{PH}, authors should compare the GPR56 expression in T_{PH} cell and non-T_{PH} cell.

We have analysed GPR56 expression on T_{PH} (PD-1^{high}) versus non T_{PH} cells (PD-1⁻) by flow cytometry in ACPA- and ACPA+ RA SF (see also comment 1a. from reviewer 1) (**Fig. 3b and below**) and show that GPR56 expression is significantly increased on PD-1^{high} T_{PH} cells. Our analysis shows that GPR56 is also expressed on T_{PH} cells in SF of ACPA- RA patients although to a lesser extent than in ACPA+ RA patients (p=0.0007).

8.Line 162, Supplementary Figure 7 is not related to the description.

Supplementary Fig. 7 is now included in a common supplementary figure for all gating strategies (as **Supplementary Fig. 1c**).

9.Line 246, the data showed no significant difference between the cytotoxic molecule expressions in CD8 T cells (Supplementary Fig. 2.), so the conclusion, cytotoxic CD8⁺ T cells also tended to be more cytotoxic in ACPA+ RA and could contribute to a perforin-dependent citrullination mechanism, is not reliable.

We agree and have removed the sentence from Discussion Section

10. Line 256, *Hobit* is one of the gene highly expressed in CD4⁺ CTL, and *Hobit* maintain the *Gzmb* expression in CD8 T cells as the reference reported. This evidence is not sufficient to conclude that '*Hobit* also drive CD4⁺ CTL differentiation in ACPA⁺ RA'.

We agree and have removed this sentence from Discussion Section

11. Current study and analysis do not give enough support for Figure 6

Based on the revised experiments, we have updated **Fig. 6f** (see below). In particular, we have removed the direct connection between CXCL13^{high} T_{PH} and cytotoxic CD4⁺ T cells and have introduced the effector CD4⁺ T cells subset as a common ancestor.

Graphical abstract

12. In supplementary Figure 2a, the representative flow cytometry dot staining of GZMB and PRF1 are same.

We apologize for this mistake. **Supp. Fig. 2a** (left panel) has been updated with the correct dot-plot.

Minor comments

1. line 85, what is the full name of SFMC?

We now provide the full name of SFMC in Page 4 Line 69.

2. Start from line 108, the process of annotating the cluster type should be more detailed.

We have detailed the annotation of each cluster page 6 (line 97-112).

3. In line 128, it should be "In PB, ADGRG1 was expressed in TPH cells".

Thank you, we have corrected this mistake.

4. Line 317, should be 1×10^6 .

This mistake has been corrected Page 18, Line 395

5. The naming of cluster 2, CD4⁺ T cells, is not rational. Because all cells are CD4⁺ T cells.

We have updated the name of all clusters (see minor comment 2). See page 6 (line 97-112).

6. *Line 452-453, The Fig 3c legend does not match the figure.*

We apologize for this mistake. Due to additional experiments included in Figure 3 on ACPA-individuals, we have removed this panel which was redundant with the new flow cytometry dot plots.

7. *Supplementary Figure 8b, the y axes are all wrong.*

We apologize for this mistake. All y axes have been renamed (CD8⁺ instead of CD4⁺)

8. *The genes mentioned from line 112 to line 121 deserve feature plots in the supplementary figure.*

We are now providing UMAP feature plots and dot plots for genes mentioned in cluster description in Supplementary Fig.5

Reviewer #3

In this manuscript by Argyriou et al., different populations of cytotoxic and peripheral helper T cell subsets are characterized by flow cytometry, single-cell RNA-sequencing and single-cell TCR sequencing in blood and synovial fluid of RA patients. The authors show that specific subpopulations of Tph and cytotoxic CD4 T cells are enriched and expanded in the SF of ACPA-positive RA patients, in specific the GPR56+ Tph population that also shows characteristics of tissue residency. The role of cytotoxic CD4 in autoimmunity is a timely topic and the extensive set of data provide important new insights in the phenotype and possible function of these cells in local inflammation. Please find comments below.

We thank reviewer 3 for her/his constructive review and interest in our study.

General comments:

One important aspect that is now missing is the relation between the identified cell subsets. In the discussion the authors postulate differentiation pathways, but they do not provide the pseudo-time data. These analysis would certainly strengthen the data and conclusions. For some of the key (functional) characteristics of the T cell subsets, protein confirmation is missing and should be added (e.g. CXCL13 and Blimp-1). Regarding the TCR sequencing data, pooling of the data may have led to a false interpretation of overlapping clone data.

We agree with reviewer 3 and have performed protein confirmation experiments, velocity analysis and patient-specific T cell clonality analysis to improve the interpretation of our findings. Please find more details in the specific comments below.

Specific comments

1- Fig 2. If I understand correctly, CD4 T cells from peripheral blood were sorted on total CD4 T cells, including naïve T cells. Since the CD4 cytotoxic and TPH populations have a memory phenotype, the proportion calculation in Fig 2D may be somewhat distorted due to the relatively high percentage of naïve CD4 T cells in PB, especially since the percentages in SF show also quite some variation. It would be insightful to also calculate the percentages of these subsets within the CD4 memory T cell fraction by excluding the naïve T cell cluster from the analysis.

We now also provide the frequencies of all clusters within the CD4⁺ memory T cell fraction (by removing the naïve CD4⁺ T cell cluster) (**Supp. Fig. 6a, Fig 2d and below**). The differences of cluster frequencies between PB and SF are still conserved when removing the naïve CD4⁺ T cell cluster. This data is presented page 7 (line 122-123).

Frequencies of CD4⁺ T cells clusters in PB and SF from n=8 RA patients (4 ACPA+, 4 ACPA-). Circle indicates ACPA+ patients, triangle indicates ACPA- patients. Line represents median, two-tailed Mann-Whitney U test.

Frequencies of CD4⁺ T cells clusters among memory CD4⁺ T cells in PB and SF from n=8 RA patients (4 ACPA+, 4 ACPA-). Circle indicates ACPA+ patients, triangle indicates ACPA- patients. Line represents median, two-tailed Mann-Whitney U test.

2- The heterogeneity of the data may be related to patient characteristics, in specific to the treatment they received. This should be discussed.

We have evaluated the frequencies of CD4⁺ T cell clusters identified in SF by single cell sequencing and flow cytometry depending on the treatment by prednisolone, methotrexate, or biologics at time of sampling (see also comment 6 from reviewer 1 and comment 2 from reviewer 2). This analysis is presented in **Supp. Fig. 7**. We did not detect any significant difference but cannot rule out the possibility that treatment will affect CD4⁺ T cell subsets in the joint. We have therefore commented this possibility in discussion page 14/15 (319-329).

3- **Fig 2**. In line 118 cluster 4 is identified as cytotoxic based on the expression of *NKG7*, *GZMH* and *PRF1*. In line 130 *HOBIT* and *GZMB* are also mentioned. It would be good to keep this consistent. *HOBIT* is also missing in the heatmap **Fig. 2c**, and *GZMB* seems to be mostly enriched in cluster 8.

We are now providing UMAP feature plots and dot plots for genes mentioned in cluster description in **Supplementary Fig. 5**

4-**Fig 2c**. ‘Heatmap showing gene expression values for a curated list of CD4 T cell subset-defining genes’; information on p-value, log-fold change, cut-off value and the source(s) the selection was based on are missing.

It would be good to provide a table with the top genes defining the different clusters (for SF and PB) in the main figures.

Thanks for this observation. We have now indicated the p-value (adjusted p-value < 0.01) and log fold change > 0.5 in **Fig 2c**. In the heatmap (**Figure 2c**), we highlight the top 3 genes and provide dot plots and UMAPs in **Supplementary Fig5a-b**. We also provide a supplementary table with differential expressed genes for each cluster with fold change and adjusted p-values (**Supp. Table 3**).

5-Line 119. Cluster 4 is also present in PB, this should be addressed.

It is true that cytotoxic CD4⁺ T cells are identified in PB with a median frequency of 5%. This is in line with previous reports showing expanded cytotoxic CD4⁺ T cells in the blood of patients with RA (Schmidt, D. *Clin Invest*, 1996; Wagner, U. *Eur J Immunol*, 2003) but also other autoimmune disorders such as myositis (Fasth, A. E. *et al. J Immunol*, 2009) and vasculitis (Moosig F, *Clin Exp Immunol*, 1998). It is currently unknown why these cells are expanded in the blood but viral antigen reactivities such as cytomegalovirus reactivity Derhovanessian, E., *J Gen Virol*, 2011) have been identified within these cells. This aspect is discussed page 15 (line 331-338).

6-Lines 120-122. Please give the frequency range also for these clusters.

In the revised **Figure 2d**, we provide the frequencies of clusters per patient as well as median values.

7-Fig 3. To show that the GPR56⁺ cells in SF differ from those in the blood it would be good to also show the (lack of) expression of PD-1, CXCL13, MAF, ICOS etc in parallel for the PB T cells.

We have analysed PD-1, CXCL13 and BLIMP-1 expression in GPR56⁻ and GPR56⁺ CD4⁺ T cells in PBMC (**Supp. Fig. 9** and **below**). Although we detected low level of CXCL13⁺ within GPR56⁺ CD4⁺ T cells, these levels were not significantly different from those obtained in GPR56⁻ CD4⁺ T cells. A slight increase in PD-1 expression was observed in GPR56⁺ CD4⁺ T cells (p=0.0260).

CXCL13, BLIMP-1 and PD-1 expression in circulating GPR56⁺ CD4⁺ T cells. Expression of CXCL13, BLIMP and PD-1 within GPR56⁻ and GPR56⁺ CD4⁺ T cells after 3 hours of CD3/CD28 activation in PBMC from n=6 RA patients. Data are from a pool of six independent experiments where a circle is a single replicate. Blue dots indicate ACPA⁻ RA and red dots indicate ACPA⁺ RA patients. Two-tailed Mann-Whitney U test.

These levels were however much lower than the levels observed on synovial GPR56⁺ T_{PH} cells (**Fig. 3d-e** and **below**). Of note, CXCL13 was detected in PB only after 3 hours of CD3/CD28 bead stimulation whereas it was largely detected in non-activated synovial CD4⁺ T_{PH} level highlighting the differences in expression.

CXCL13 and BLIMP-1 expression within PD-1-GPR56- (non-TPH), PD-1high GPR56low and PD-1high GPR56+ (2 TPH states) in CD4+ T cells in ACPA+ and ACPA- RA synovial fluid, quantified in n=5 ACPA- SF and n=5 ACPA+ SF

This result is confirmed by the sc data set where the frequencies of CXCL13^{high} T_{PH} were very low in blood (**Figure 2d**, median frequency of 0.2%). Altogether, we can conclude that GPR56 identifies T_{PH} cells in SF and mainly identifies cytotoxic CD4⁺ T cells in blood. We cannot rule out the possibility that T_{PH} expressing GPR56⁺ CD4⁺ T cells are also present in blood, especially at earlier time point in the disease and GPR56 would be therefore a good marker to follow also in peripheral blood. This data is presented page 7 (line 139-140).

8-Fig 3b. Is there a significant difference between ACPA+ and ACPA- regarding the expression of PD-1 and MHCII if you compare all individuals?

Both PD-1 and GPR56 expression (MFI) were increased on CD4⁺ T cells in SF of ACPA+ RA patients whereas MHCII expression was not significantly different. This set of data is presented in **Fig 3b, Supp Fig. 12** and below.

PD1, MHCII and GPR56 expression on CD4⁺ T cells in ACPA+ and ACPA- RA SF. PD1, MHCII and GPR56 MFI (mean fluorescence intensity) on CD4⁺ T cells in ACPA- and ACPA+ RA SF (n=9 ACPA- SF and n=11-12 ACPA+ SF). Data are from a pool of nine independent experiments where a circle is a single replicate. Line indicates median, two-tailed Mann-Whitney U test. ns: not significant. Blue dots indicate ACPA- RA SF and red dots indicate ACPA+ RA SF.

9-Fig 4. Since *CXCL13* and *PDRM1* are defining the TPH clusters, this should also be confirmed on the protein level (co-expression with *GPR56*) as is shown for other markers.

We agree and we took advantage of the differential *GPR56* expression on the two T_{PH} subsets (**Fig. 3a, c** and **Supp. Fig. 11a**) to evaluate the expression of MHCII, PD-1, *CXCL13* and BLIMP-1 in ACPA- and ACPA+ RA patients (see also comment 1b from reviewer 1). We confirmed that PD-1^{high} *GPR56*⁺ T_{PH} are increased in ACPA⁺ SF (**See below and Fig. 3c**). We observed that, PD-1^{high} *GPR56*⁺ T_{PH} are indeed producing higher *CXCL13* and express higher levels of MHCII and PD-1 than the PD-1^{high} *GPR56*^{low} T_{PH} . However, PD-1^{high} *GPR56*⁺ T_{PH} also expressed higher level of BLIMP-1 (encoded by *PRDM1*) (**Figure 3e and below**). Based on the discrepancies between *PRDM1* and BLIMP-1 expression, we have therefore renamed these subsets: *CXCL13*^{low} T_{PH} (previously *PRDM1* T_{PH}) and *CXCL13*^{high} T_{PH} (previously *CXCL13*⁺ T_{PH}). All other proteins tested (*CXCL13*, PD-1, MHCII, *GPR56*, *LAG3*) correlated with the single cell RNA sequencing data. (See result section page 8, line 157-171 and discussion section page 14, line 309-319).

Representative flow cytometry dot plots of the two T_{PH} states (PD-1^{high} *GPR56*^{low} and PD-1^{high} *GPR56*⁺) and quantification of their frequency, MHCII and PD-1 expression in n=9 ACPA- SF and n=11-12 ACPA+ SF. *CXCL13* and BLIMP-1 expression within PD-1-*GPR56*⁻ (non-TPH) and PD-1^{high} *GPR56*^{low} and PD-1^{high} *GPR56*⁺ (2 T_{PH} states) in CD4⁺ T cells in ACPA- and ACPA+ RA SF in n=5 ACPA- SF and n=5 ACPA+ SF. Line represents median, two-tailed Mann-Whitney U test

10-Line 159. The authors conclude that the cytotoxic CD4 T cells are recirculating and the TPH not. Nevertheless TPH cells are also found in circulation and also show some overlap in TCR. So although these cells do not express tissue resident markers, there may be (re)circulating.

We agree that this conclusion is overstated since we don't have experimental evidence showing that T_{PH} cells are not re-circulating. We have therefore removed this sentence from the discussion and soften our discussion as stated :” Overlap clonality between cytotoxic clones in PB and SF suggest their recirculation”, page 15, line 333-336

11-Fig 4c. When addressing tissue residency, *CD69* expression is indeed one of the hallmarks. Therefore it would be informative to not only show the MFI but also the percentage expressing cells within the populations.

Based on the reviewer's comment, we have now included the percentage of *CD69*⁺ cells in **Fig. 4b-c** (CD4⁺ T cells, **see below**) and in **Supp Fig. 14** (CD8⁺ T cells). We show that the frequency of *CD69* is also increased within the *GPR56*⁺ CD4⁺ T cell subset. For consistency, we have also included the MFI for all other tissue-resident memory markers tested (**Fig. 4b-c and Supp Fig.14**) which also show significant differences.

Expression of LAG-3, CXCR6, CD69, CD49a and CX3CR1 on GPR56+ and GPR56- CD4⁺ T cells in ACPA+ RA, in n=10 ACPA+, n=8 ACPA-, LAG3, CXCR6, CD69, CD49a and n=8 ACPA+, n=7 ACPA-, CX3CR1). White dots indicate GPR56- CD4⁺ T cells and black dots indicate GPR56⁺ CD4⁺ T cells. Line represents median, two-tailed Mann-Whitney U test. Data are from a pool of eight independent experiments where a circle is a single replicate.

12-TCR data. *In what percentage of T cells a TCR beta and/or alpha chain could be detected? In figure 5a, 7 out of 10 clones represent the beta chain. Is this because of a difference in expansion or a difference in sequence recovery? Also, with 58bp the reads are rather short. Very low efficiency may introduce bias in the data, please comment.*

We have performed new single cell experiments on CD4⁺ T cells from PB and SF from ACPA+ (n=4) and ACPA- (n=4) RA (see question 6 from reviewer 2 and question 1 reviewer 1). Libraries for RA8-15 on which we do the TCR analysis were sequenced on Illumina NextSeq500/550 high output 150-cycle v2.5 kits with the following read structure: read1: 26, read2: 90, index 1: 8, index 2: 8 as per 10X genomics recommendation. We present TCR efficiency recovery in **Supp. Table 6a**. Sample 12 (ACPA-) was excluded because of low TCR recovery (below 2%) as indicated in the text page 10, line 214-215. In the other 7 samples, we detect paired alpha/beta chains with a median frequency of 66% in SF and 62% in PB (efficiency per patient and compartment is included in **supplementary Table 6a**). The raw TCR data per patient and compartment is included in **supplementary table 4 and 5**. For downstream TCR analysis we exclusively analysed paired alpha/beta chains clones.

12-Fig 5b. *The most expanded clones seem to be present within the cytotoxic CD4 T cell population, although two patients (RA#2 and RA#7) show a completely different pattern. It would be good to mention this in the text.*

We have run new single cell experiments on PB and SF samples from 4 ACPA- and ACPA+ RA patients. Although we detect expanded CD4⁺ T cell clones within the cytotoxic CD4⁺ T cell population, most of the SF expanded clones are present within the CXCL13^{high} (GPR56+) T_{PH} cell cluster (**Fig 5b** and **below**). In ACPA- SF samples, two patients presented with expanded Tregs. In blood, we observe a different profile with most expanded clones found within cytotoxic and central memory CD4⁺ T cell cluster independently of the ACPA profile. The difference of clonality observed between the first and second set of experiments might originate from treatment or patient-specific differences. Indeed, in the first set of experiments,

RA#2 and RA#7 SF samples which presented with expanded CXCL13⁺ T_{PH} cells were HLA-shared-epitope positive which is a common characteristic of all patients included in the second set of experiments. Treatment might also affect clonal expansion and is discussed page 15 (line 321-329).

Stacked barplots displaying the phenotype of the SF (left) and PB (right) expanded clones (n ≥ 2 cells) in each RA patient, quadrant represents individual clone. c) Frequency of expanded clones within each CD4⁺ T cell clusters in SF (left panel) and PB (right panel) for each RA patient (n=4 ACPA+, 3 ACPA-).

14-Fig 5c-d. If I understand correctly, in these figures all sequences from different patients are pooled. Since the full HLA background of the patients is unknown, it does not seem to be fair to draw conclusions about shared clones between subsets and between PB-SF. This would be more relevant on the individual level (so is there TCR overlap between subsets or between PB and SF within a patient). For the level of overlap it would be good to mention percentages.

We agree with this comment (see also comment 3 from reviewer 1). In the revised **Figure 6b** and **below b**), we are now presenting shared CDR3 sequences between all cells (including clones) in the different clusters. The CDR3 overlap is quantified using a jaccard index. Moreover, we also analysed the sharing of CDR3 clones by focusing on expanded clones (n ≥ 2) in each cluster per patient (**Figure 6c** and **below c**). Finally, we provide raw information on shared CDR3 in expanded clones in **supplementary table 7**). In SF, CXCL13^{high} T_{PH} cells show clonal overlap with CXCL13^{low} T_{PH} (Jaccard Index (JI):0.038), proliferating (JI:0.031), effector (JI:0.023) and cytotoxic CD4⁺ T cells (JI:0.013). We then focused our analysis on expanded clones in CXCL13^{high} T_{PH} CD4⁺ T cells (n ≥ 2). A median of 59% of the CXCL13^{high} T_{PH} expanded clones shared CDR3 sequences with cells from other clusters (either clones or unique cells) (**Supp. Table 6d**). These shared sequences were identified in all patients. 28% of these clones were shared with proliferating CD4⁺ T cells, 25% with CXCL13^{low} T_{PH}, 19% with effector CD4⁺ T cells, and 6% with cytotoxic CD4⁺ T cells (**Fig. 6c, Suppl. Table 6e**).

Altogether, we confirmed a sharing of CDR3 sequences between CXCL13^{high} T_{PH} expanded clones and CXCL13^{low} T_{PH} cells (clones or unique cells) as well as effector CD4⁺ T cells and to a lesser extend cytotoxic CD4⁺ T cells. This data is presented page 11-12 and discussed page 14 (line 301-303).

b) Jaccard overlap quantifications for clonotypes between cell clusters across compartments. **c)** % of TCR clonality overlap between expanded T cell clones (n ≥ 2 cells) and T cells from other clusters (n ≥ 1 cell).

15-Lines 181-183. and supplementary Fig 9. The authors state that identical clones are shared between individuals. However, only the beta chain sequence is shown to be overlapping and therefore the TCR may be different between these individuals. Also, why was a cut-off of 4 used in this setting?

We agree with this comment and have focused our analysis on paired alpha/beta TCRs. In the analysed dataset, we don't find any shared sequences between individuals. A cut-off of 4 had been applied to facilitate visualization but we agree that for consistency, it is better to visualize all clones with the define cut-off of 2.

16-Discussion. The authors postulate that the TPH clusters correspond to two states of differentiation (lines 222-225) and that TPH cells lead to cytotoxic CD4 T cell expanded clones (lines 274-276). By performing a pseudotime analysis, the authors can demonstrate whether this is indeed the most likely connection. This would further strengthen the data.

We totally agree with this suggestion (see also comment 4 from reviewer 1 and comment 4 from reviewer 2). To evaluate the conversion from CXCL13^{low} T_{PH} into CXCL13^{high} T_{PH}, we sorted GPR56^{low} PD-1^{high} CD4⁺ T cells and GPR56^{high} PD-1^{high} CD4⁺ T cells from ACPA+ RA SF and submitted these cells to 48 hours CD3/CD28 bead activation. We then evaluated the expression of PD-1, GPR56, CXCL13, BLIMP-1 and Perforin-1 (**Figure 6d and below**). We observed that GPR56^{low} T_{PH} cells upregulated PD-1 to a level comparable to CXCL13^{high} T_{PH} (p=0.0005). We also observed an upregulation of GPR56 and BLIMP-1 in all 3 patients tested which did not reach significance due to lack of power. Although our initial plan was to perform n=6 experiments, we could not recover enough PD-1^{high} GPR56^{low} T_{PH} cells in 3 of the tested samples. Finally, we did not observe any striking upregulation of perforin-1 expression. Altogether, these data shows that CXCL13^{low} T_{PH} can convert into CXCL13^{high} T_{PH} but not in cytotoxic CD4⁺ T cells in the tested conditions. This data is presented page 12 (line 258-265) and discussed page 14 (line 303-308). Of note, we didn't see any increase in CXCL-13 production in the 2 T_{PH} subsets after T cell activation which might be due to the lack of additional stimulation factors as discussed page 12 line 263-265.

Activation of T_{PH} subsets. Representative flow cytometry dot plots showing the expression of PD-1, GPR56, CXCL13, BLIMP-1, and PERF-1 in control (upper panel) or 48hours CD3/CD28 activated T_{PH} cell states (lower panel) (PD-1^{high} GPR56^{low} and PD-1^{high} GPR56⁺ from ACPA+SF), quantified in n=3 ACPA+ RA patients. (d) Data are from a pool of three independent experiments where a circle is a single replicate. Line indicates median, two-tailed Wilcoxon paired test.

Further, to explore the connection between CD4⁺ T cell subsets (see also comment 4 from reviewer 1 and comment 4 from reviewer 2), we performed velocity analysis on the sc dataset (**Figure 6e** and **below**). We indeed observe a transition from CXCL13^{low} T_{PH} into CXCL13^{high} T_{PH} which was more obvious in ACPA+ CD4⁺ T cells (lower panel). Interestingly, effector CD4⁺ T cells and SESN3⁺ CD4⁺ T cells project into two independent branches leading to T_{PH} in one direction and cytotoxic CD4⁺ T cells in the other direction. These observations suggest that, in RA, T_{PH} and cytotoxic CD4⁺ T cells are differentiating from effector T cells through two independent pathways. The sharing of CDR3 sequences that we observe between these two subsets could be due to common progenitors present in the effector CD4⁺ T cell subset populations. This data is presented page 12 (line 265-276) and discussed in page 14 (Line 304-308), page 15 (Line 344), page 16 (345-351).

Velocity plot showing the connection between the different CD4⁺ T cell clusters in RA split in ACPA- (upper panel, n=4) and ACPA+ (lower panel, n=11).

17-Minor comments

Please use T-cell or T cell consistently

We now used “T cell” throughout the manuscript

18-Line 160. this statement refers to SF, please add.

Thank you for notifying this omission. We have modified the sentence to: “A similar tendency was also observed for the expression of *CXCR6* in SF.” page 9, Line 193.

REVIEWERS' COMMENTS

Reviewer #1 (Remarks to the Author):

The revised manuscript by Argyriou et al has addressed my concerns. I thank the authors for performing such a comprehensive review, particularly concerning the TCR sequences and the RNA velocity analysis, to clarify the interrelation among the different CD4 subsets.

Minor points:

1. As a result of additions/modifications, the abstract has lost in structure and readability. It should be revised, especially the sentences marked:

Lines 7,8:

Unclear meaning.. two distinct TPH states characterized by the differential expression of CXCL13?

Line 9: circulating cytotoxic CD4+ cells?

Lines 12,13,14: sentence is difficult

Line 15: confined refers mostly to space... contained?

2. Line 123: "Due to low sample size, no significant differences were observed between ACPA- and ACPA+ patients"

There is no guarantee that significance would be achieved with more samples.

Reviewer #2 (Remarks to the Author):

The authors have addressed all my questions. Some minor points:

1. The full name of ACPA should be indicated in the Abstract.
2. Line 189-191, LAG-3 is not mentioned in ref 30. Moreover, LAG-3 is upregulated in some Trm cells, while it is not appropriate to characterize LAG-3 as a resident memory marker.
3. Figure 4b, the proportions of CD69+ may not match the dot plots.

Reviewer #3 (Remarks to the Author):

The authors have addressed all the issues that I had raised. The conclusions are now better substantiated with new data, including data on the relation between the identified subsets and protein expression data. I have no further comments.

Sincerely,

Femke van Wijk

Response to reviewer comments:

Reviewer #1 (Remarks to the Author):

The revised manuscript by Argyriou et al has addressed my concerns. I thank the authors for performing such a comprehensive review, particularly concerning the TCR sequences and the RNA velocity analysis, to clarify the interrelation among the different CD4 subsets.

We thank reviewer 1 for her/his thorough examination of our revised manuscript.

Minor points:

1. As a result of additions/modifications, the abstract has lost in structure and readability. It should be revised, especially the sentences marked:

Lines 7,8:

Unclear meaning.. two distinct TPH states characterized by the differential expression of CXCL13?

Line 9: circulating cytotoxic CD4+ cells?

Lines 12,13,14: sentence is difficult

Line 15: confined refers mostly to space... contained?

We have revised the abstract to 147 words and have revised these unclear sentences, thanks for pointing that out.

2. Line 123: “Due to low sample size, no significant differences were observed between ACPA- and ACPA+ patients”

There is no guarantee that significancy would be achieved with more samples.

We agree and have removed “due to low sample size” from this sentence.

Reviewer #2 (Remarks to the Author):

The authors have addressed all my questions.

We are grateful for reviewer 2´s thorough examination of our revised manuscript.

Some minor points:

1. The full name of ACPA should be indicated in the Abstract.

We have now included the full name of ACPA in the abstract.

2. Line 189-191, LAG-3 is not mentioned in ref 30. Moreover, LAG-3 is upregulated in some Trm cells, while it is not appropriate to characterize LAG-3 as a resident memory marker.

We agree and have removed LAG-3 from all sentences describing tissue resident memory receptors. We instead introduce LAG-3 as an inhibitory receptor. We added a new sentence and reference page 9 line 349/350 (manuscript with track changes) to justify the inclusion of

LAG-3 in this study: “We also evaluated the expression of the inhibitory receptor *LAG-3* which is often co-expressed with *PD-1* on tumor infiltrating lymphocytes³².”

2. Figure 4b, the proportions of CD69+ may not match the dot plots.
We apologize for this mistake which is now corrected.

Reviewer #3 (Remarks to the Author):

The authors have addressed all the issues that I had raised. The conclusions are now better substantiated with new data, including data on the relation between the identified subsets and protein expression data. I have no further comments.

We are grateful for reviewer 3 ´s thorough examination of our revised manuscript.